



# Flood type classification and assessment of their past changes across Europe

Yeshewatesfa Hundecha[1], Juraj Parajka[2], Alberto Viglione[2]

[1]Swedish Meteorological and Hydrological Institute, Folkborgsvägen 17, 601 76, Norrköping, Sweden

[2]Institute of Hydraulic Engineering and Water Resources Management, Vienna University of Technology, Karlsplatz 13 E222/2, 1040, Vienna, Austria.

*Correspondence to*: Yeshewatesfa Hundecha (yeshewatesfa.hundecha@smhi.se)

**Abstract.** This study was carried out to establish the characteristics of observed flood events across Europe in the past in terms of their spatial extent and the processes leading up to the events. Daily discharge data from more than 745 stations of the Global Runoff Data Centre were used to identify peak flows at each station for the period 1961-2010. The identified events at the different stations were further analysed to determine whether they form the same flood event, thereby delineating the spatial extent of the flood events. A pan-European hydrological model was employed to estimate a set of catchment hydrological and hydro-meteorological state variables that are relevant in the flood generation process for each of the identified spatially delineated flood events. A subsequent clustering of the events based on the simulated state variables was used to identify the flood generation mechanism of each flood event. Four general flood generation mechanisms were identified: long-rain flood, short-rain flood, snowmelt flood, and rain-on-snow flood. A trend analysis was performed to investigate how the frequency of each of the flood types has changed in time over the investigation period. In order to investigate whether there is a regional and seasonal pattern in the dominant flood generating mechanisms, this analysis was performed separately for winter and summer seasons and five different regions of Europe: Northern, Western, Eastern, Southern Europe, and the Alps. Continentally, the total number of flood events didn't show a significant change. However, the frequency of winter long rain events increased significantly while that of summer rain-on-snow events decreased significantly over the investigation period. Regional differences were detected in the dominant flood generating mechanism and the corresponding trends. In Western Europe, the frequency of both winter and summer rainfall events increased significantly. In Northern and Eastern Europe, the frequency of summer rain-on-snow events decreased significantly. In addition, winter short rainfall events increased significantly in Eastern Europe. In the Alps, the frequency of summer short rain events increased significantly.



## 1 Introduction

The frequent occurrence of extreme flood events in different parts of Europe in the recent past has raised interest in scrutinizing and understanding the underlying causes (Ulbrich et al., 2003a, b; Marsh 2008; Blöschl et al., 2013, 2016;

Schröter et al., 2015). The studies analyzed the meteorological or both meteorological and hydrological conditions that led to the events. There has also been increasing interest in assessing whether there has been an increase both in the frequency and magnitude of flooding in Europe over the past decades and whether there is a likelihood of their increase in the future. Mediero et al. (2015), for instance, studied past changes in the magnitude and frequency, as well as timing of flooding across Europe using discharge data collected from different sources. A number of other studies have also been carried out for

different parts of Europe (See Hall et al., 2014 for a review of studies on flood changes across Europe).

Understanding the dominant process controls of flooding and the main drivers of detected changes in flooding is a key to proper flood risk management since the important controls vary depending on local conditions (Kundzewicz et al., 2014; Blöschl et al.,2015). The reliability of estimation and analyses of floods can also be enhanced if information on process controls is incorporated into the analyses (Loukas et al. 2000, Merz and Blöschl, 2008; Rogger et al. 2012). Several studies

have been carried out to find out the meteorological and hydrological process controls of flood generation at different scales. Nied et al (2013) and Rogger et al (2013), for instance, investigated the role of catchment soil moisture in the generation of floods. Sui and Koehler (2001) analyzed the importance of snowmelt and rain-on-snow for the generation of peak flows in the upper part of the Danube basin using concurrent measurements of precipitation, snow water equivalent and discharge. Similarly, McCabe et al (2007) investigated the importance of rain-on-snow on flood generation in the western US. The

importance of synoptic atmospheric processes was studied by Parajka et al (2010) and Prudhomme and Genevier (2011), among others.

There is evidence of changes in flood regime in different parts of Europe over the past decades, albeit in a regionally different pattern (Hall et al., 2014; Blöschl et al., 2015). Different studies have attempted to attribute the changes to their potential drivers for different parts of Europe. Hattermann et al (2013) attributed changes in flooding in Germany to changes

in air temperature and precipitation, as well as flood prone large scale circulation patterns. Pinter et al (2006) attributed increases in flooding in the Rhine to increased precipitation and a change in landuse. Prosdocimi et al (2015) investigated the relative importance of precipitation and urbanization to two UK catchments. Van der Ploeg and Schweigert (2001) attributed increased flooding along the Elbe River to the intensification of agriculture.

Most of the flood change attribution studies rely on comparing observed changes in certain statistical characteristics of the

potential drivers to the detected changes in flooding and hypothesizing that the change in the potential driver is responsible for the detected change in flooding (Merz et al., 2012). There are, however, some studies that employed models to investigate whether changes in the hypothesized potential drivers translate into a change in flooding. For instance, Hundecha



and Merz (2012) investigated the relative importance of precipitation and temperature in explaining changes in flooding for different catchments across Germany by introducing the year to year variability in one of the meteorological variables used to drive a hydrological model while keeping the other stationary. Similarly, Vorogushyn and Merz (2013) investigated the role of river training works on the change in flooding along the Rhine by comparing model simulations under consideration

of the river training works with homogenized flows. Along the Danube, Skublics et al. (2016) investigated the effect of river training on the flood retention characteristics by systematic two-dimensional hydrodynamic modelling. They found that extreme floods are attenuated more strongly in the present state of the channel-flood plain system than they were historically. Employing model simulations to link the potential drivers of flood change to the detected changes would allow framing the attribution exercise based on a direct assessment of the changes in the flood generation processes. Changes in flooding might

be attributed to multiple drivers, such as climatic drivers, landuse, and river training. In a classical statistical framework of attribution, it could be difficult to separate the effects of the different potential drivers on the change in flooding (Merz et al., 2013). However, some studies have attempted to assess the relative importance of different drivers through investigation of the significance of covariates related to the different drivers used in statistical distributions of flood flows (Prosdocimi et al., 2015; Šraj et al., 2016, Viglione et al. 2016). Employing a simulation based assessment would enable identifying the relative

importance of the different drivers in explaining the detected changes in flooding if enough information were available on the temporal evolution of the different drivers.

One way of assessing changes in flood regime and attributing the changes would be to classify flood events based on their generating mechanisms and to analyse the changes in the occurrence of the different types of flood events in time. Several studies were conducted in the past to classify flood events based on their dominant generating mechanisms. Loukas et al

(2000) employed a hydrological model to simulate different components of the annual peak and peaks over threshold to investigate the flood generating mechanisms to two Canadian catchments. Merz and Blöschl (2003) classified flood events across Austria based on climate inputs (rainfall, snowmelt) and basin states (soil moisture, snow cover). Sikorska et al (2015) classified flood events in mountainous Swiss catchments using characteristics of precipitation and catchment states, such as catchment wetness, snow cover and glacier cover. Turkington et al (2016) classified synthetic flood events generated

through application of a weather generator together with a hydrological model by using a cluster analysis technique on a set of meteorological indices derived from the generated synthetic weather for two Alpine catchments in France and Austria.

Most of the previous flood type classification studies focused on either catchment or national scale classification of events. However, a large scale regional assessment offers a regional pattern of flood risk, which can potentially facilitate coordination of regional flood risk management, as is emphasized by The European Flood Directive (EU, 2007). There is,

therefore, an increasing importance placed on a regional research effort. A recent contribution to this kind of effort is the work by Berghuijs et al (2016), who classified annual maximum flows across the US by assessing the correspondence of seasonality of the events with that of the different descriptors used to define flood event types. They considered extreme precipitation, soil moisture excess precipitation, snowmelt and rain-on-snow processes for their classification.



The aim of the present study is to establish different flood process types and evaluate their clustering in space and time across Europe. The idea is to explore the relative role of rainfall processes, snow melting and river basin state of soil moisture to identify the flood type genesis across Europe and to assess past changes in the dominant flood generating mechanisms in different regions of Europe. We make use of a consistent daily discharge data set across Europe to select

flood events and derive the corresponding meteorological forcing from reanalysis data. Furthermore, we employ a regional hydrological model to estimate the corresponding state variables, such as soil moisture deficit, snow water equivalent and snowmelt. We group the selected flood events in terms of their dominant generating mechanism based on the corresponding meteorological forcing and state variables. Finally, we assess the regional pattern of the dominant flood generating mechanisms and the presence of any temporal trend in the pattern.

**2 Data and Method**

**2.1 Data**

We based the identification of flood types and assessment of their changes across Europe on pan-European open datasets of discharge, physiographic and climate/meteorological characteristics. We made use of a global dataset of daily discharge time series, available at Global Runoff Data Centre (GRDC, http://www.bafg.de/GRDC/EN/Home/homepage_node.html)

from 747 stations across Europe. The data period for the stations is variable, with an average length of 54 years. After screening the stations for coverage of data over the investigation period (1961-2010), 614 stations were kept for further analysis (see Fig. 1). A hydrological model was employed for the simulation of hydrological state variables that are used for flood type classification. Different open data sets were used to set up the hydrological model (Hundecha et al., 2016). River networks and subcatchments were delineated using WWF's Hydrosheds data (Lehner et al. 2008) for the model domain

south of 60 degrees latitude and from Hydro1K (Verdin, 1997) further north. Hydrological response units (HRUs) were derived from landuse and soil data obtained from different sources. Landuse was derived from the CORINE landuse data and GlobCover data (Arino et al. 2008) where CORINE does not have coverage. Lakes and reservoirs were extracted from GLWD (Lehner and Döll, 2004) and GranD (Lehner et al. 2011) data sets respectively. Irrigated areas were identified from GMIA (Siebert et al. 2010) and MIRCA (Portmann et al. 2010) data sets. Soil types were derived from the European Soil

Database, ESDB (Panagos, 2006) and Digital Soil Map of the World (DSMW) data sets. The WATCH and WFDEI (Weedon et al. 2014) meteorological forcing data over the period 1901-2010 were also used to simulate hydrological variables using the employed hydrological model. The WATCH data was used as forcing for the period 1901-1978 and WFDEI forcing was used for the period afterwards.




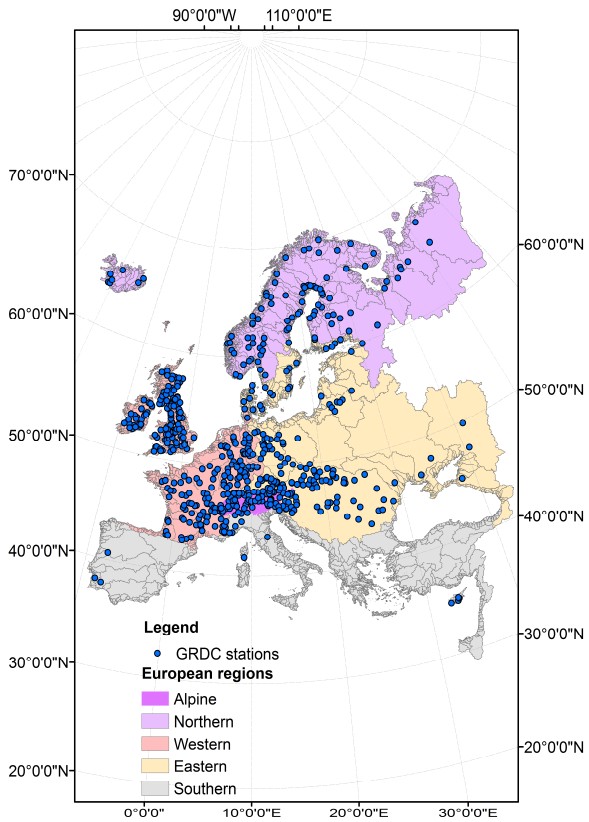

**Figure 1**: Hydro-climatological regions of Europe and distribution of GRDC stations used in the study.

**2.2 Selection of extreme flow events**

5    Independent flood events and the corresponding peak flows were identified at each gauging station from the GRDC daily
discharge time series as a basis of flood type classification. Events were identified by employing a base flow separation
technique. A traditional procedure for base flow separation starts with identification of the points at which the direct runoff
starts and ends. The start point is readily identified as the time when the flow starts to increase, while the end-point is usually
taken as the time when a plot of the logarithmic transformed discharge values against time becomes a straight line. A wide

10    range of techniques is available for establishing the end-points for separating the base flow from the direct runoff (see e.g.



Chapman, 1999). In this experiment we applied the Chapman digital filter (Chapman, 1999), which estimates base flow as a simple weighted average of the direct runoff and the base flow at the previous time interval, i.e.

$$Q_b(i) = kQ_b(i-1) + (1-k)Q_d(i) \qquad (1)$$

where $Q_b(i)$ and $Q_d(i)$ are the base flow and direct runoff, respectively, at time interval $i$ and the parameter $k$ is the recession

constant during periods of no direct runoff. If a given discharge $Q(i)$ represents the sum of base flow $Q_b(i)$ and direct runoff $Q_d(i)$, then $Q_b(i)$ can be estimated as:

$$Q_b(i) = \frac{k}{2-k}Q_b(i-1) + \frac{1-k}{2-k}Q(i) \qquad (2)$$

Estimation of the recession constant $k$ for each catchment follows the approach of Vogel and Kroll (1996) (see also Thomas et al., 2013). The approach consists of the following steps:

1) identification of the start of discharge recession when a 3 day moving average begins to decrease and the end when a 3 day moving average begins to increase;

2) selection of recessions with length larger than or equal to 10 days;

3) removal of the first three points of the recession to eliminate effects of averaging;

4) fitting the model $ln(Q) = ln(Q_0) + ln(k)*t +$ error for each recession using the ordinary-least-squares method, i.e., estimate

$k$ for individual events; and

5) estimation of the mean recession constant k of all the analysed recessions.

Once the recession constant is estimated, time-series of $Q_b(i)$ and $Q_d(i)$ are computed and independent discharge events are separated. The flood event peaks are then represented by the maximum daily discharge within each independent event if the direct runoff is greater than both the base flow and mean annual direct runoff. This criterion is introduced to eliminate cases

where discharge or base flow equals zero.

**2.3 Hydrological and hydrometeorological state variables**

A semi-distributed continuous daily rainfall-runoff model that was setup for the entire Europe, E-HYPE (Donnelly et al., 2016; Hundecha et al., 2016), was employed to estimate a set of hydrological state variables. The model domain covers an area of 8.8 million km$^2$ and is subdivided into 35,408 subcatchments with an average size of 248 km$^2$. Each subcatchment is

further subdivided into hydrological response units (HRUs) based on a combination of different landuse classes and soil types. The model has conceptual routines for the major land surface and subsurface processes. The snow accumulation and





melt process is modelled using the degree-day method. Rainfall and/or snowmelt are apportioned into surface and subsurface flow components using different soil and landuse dependent thresholds and parameters. Potential evapotranspiration (PET) is estimated using the modified Jensen-Haise model (Oudin et al., 2005). PET is achieved only if the actual soil moisture exceeds a certain threshold and actual evapotranspiration decreases linearly from the PET value at this threshold to zero at

wilting point. The generated runoff is routed through each subcatchment and between subcatchments using a simple river routing routine which simulates attenuation and delay within the river system. A simple routing procedure for flows out of lakes and reservoirs is also employed. The model is driven by the WATCH and WFDEI forcing data sets. These data sets provide daily gridded precipitation and air temperature with an approximate grid size of 50 km. The data are interpolated to the centroid of each subcatchment in the model domain. The WATCH data set was used for the period 1961-1978 while the

WFDEI data set was used for the subsequent period.

The model parameters were estimated as functions of catchment physiographic attributes that control the processes they describe through calibration against observed daily discharge at a set of gauging stations. To account for the possible variation of the relationships, the model domain was subdivided into different classes based on a set of catchment physiographic and climate attributes and the model parameters were regionalized separately within each class. Some of the

parameters were estimated against satellite based observations. The landuse dependent PET parameters were calibrated against the MODIS global data set. The landuse dependent snow accumulation and melt parameters, which are of particular interest for the present work (see Table 1 and Section 2.5), were estimated against the GlobSnow snow water equivalent data. The regionalized model was evaluated at more than 500 independent validation discharge stations across Europe. In addition to evaluating the model's skill in simulating daily discharge series using the standard metrics such as the Nash-

Sutcliffe efficiency measure and model bias, evaluation of the model's ability to simulate different flow signatures, including high and low flows as well as flow variability, was performed. Overall, the model performs reasonably well. Details of the model setup, the data used and the calibration and validation procedure are presented in Hundecha et al. (2016).

For each flood event identified at each discharge station, the corresponding meteorological data were derived from the forcing data used in the hydrological model as catchment average values. Total precipitation amount corresponding to the

event was computed as the total amount between the event start and end dates. Furthermore, antecedent precipitation index corresponding to different lengths of days before the event start date were computed using the catchment average daily precipitation. Hydrological state variables at the onset and during each flood event were estimated from the E-HYPE model simulation. Notwithstanding the known uncertainty in model simulated variables, based on the results of the rigorous model validation discussed above and implementation of additional earth observation data to constrain some of the parameters that

are most relevant for the present work, we assume that the model simulated variables can reasonably be used for the present work at the scale the work focuses on. Table 1 shows the hydrometeorological and hydrological variables considered for the flood type identification.





**Table 1**: Hydrological and hydro-meteorological state variables used for characterizing and classifying flood events.

| Variable | Description | Source |
|---|---|---|
| Precipitation [mm] | Basin average precipitation amount between event onset and flood peak date | WFDEI, WATCH |
| Rainfall fraction [-] | Proportion of  basin average precipitation amount falling as rainfall | E-HYPE |
| Snowfall fraction [-] | Proportion of  basin average precipitation falling as snow | E-HYPE |
| Snow WEQ [mm] | Average basin accumulated water equivalent of snow at the onset of event | E-HYPE |
| Smelt [mm] | Basin average snowmelt amount between event onset and flood peak date | E-HYPE |
| SW deficit [mm] | Average basin deficit of soil moisture to reach field capacity at event onset | E-HYPE |
| API (n) [mm] | Antecedent precipitation index corresponding to n days prior to event onset | WFDEI, WATCH |

**2.4 Spatial clustering of flood events**

The events identified at individual gauging stations were clustered in space to evaluate the spatial extents of individual flood

events. This enables the identification of the spatial extents of coherent flood events. Spatial clustering of events was performed by analysing the dates of the peak flows at the individual gauges and the distance between the centroids of the catchments draining to the stations. If peaks at two stations occur within a short time interval and the catchments draining to the stations are also close to one another, it is likely that the peaks at the two stations belong to the same event. The maximum time interval between the peaks and the corresponding maximum distance between the basins for peaks to be of

the same event depends on a number of factors, such as the topological configuration of the catchments, the characteristics of the storms or meteorological conditions that lead to the events, as well as the magnitude of the flood. Therefore, some subjectivity is ultimately involved in spatially clustering events in this way. For instance, Merz and Blöschl (2003) assumed annual maximum flows occurring within a time lag of 1 day to be of the same event if the centroids of the contributing catchments are within 50 km. In order to allow for catchment reaction time and flood propagation time downstream along

bigger rivers, Uhlemann et al. (2010) employed a time window of 3 days ahead and 10 days following a peak discharge of at least a 10 years return period to group flood events at multiple locations as the same event.

In the present work, a spatially coherent flood event is defined as an event where at least one gauge records a peak flow of 5-years flood or larger and all the others that are grouped into one event record at least a 2-years flood. Catchments whose centroids lie within 50 km from one another while their peak dates lag by a maximum of one day are assumed to be of the

same event. Furthermore, if the gauging stations are connected, a peak that occurs at the downstream gauge within 3 days is considered to belong to the same event. Depending on the basin size and the distance between the gauges, this might be longer. For instance, Uhlemann et al. (2010) found a time lag of 8 to 10 days in the Elbe river basin from Dresden to Neu



Darchau. In the present work, there are several intermediate gauging stations between the most upstream and downstream gauges within large basins and the definition of 3 days lag is considered reasonable.

Since the season in which a flood event occurs can also offer additional information on the likelihood of a given flood triggering mechanism, the identified spatially clustered flood events were separately analyzed for the hydrological winter

(October – March) and summer (April – September) seasons.

## 2.5 Classification of flood events based on their generating mechanisms

Four types of flood events were defined based on the hydro-meteorological conditions and the catchment states that resulted in them: short-rain floods, long-rain floods, snowmelt floods, and rain-on snow floods.

*Short-rain floods*: are events that result from intense rainfall with a duration of a few hours. In this work, daily

meteorological data are used because of the large spatial coverage of the study area. Therefore, short-rain flood is defined as a flood event caused by rainfall of duration less than or equal to a day.

*Long-rain floods*: are events triggered by rainfall with duration of several days. The intensity could be low but may gradually saturate the catchment and may ultimately result in flooding. In this work, long-rain flood is defined as an event resulting from rainfall of duration more than a day.

*Snowmelt floods*: occur when there is an accumulated snow in the catchment and the temperature rises above a freezing point. In this work, an event is considered as a snowmelt flood event if the model simulation yields snowmelt while there is little precipitation between the flood onset and flood peak.

*Rain-on-snow floods*: The snowmelt process may be enhanced during rainfall due to the additional latent heat the rain provides to the snowpack. Together with the incoming rainfall, the snowmelt can result in considerable runoff. An event is

defined as rain-on snow flood in this work if the model simulates snowmelt and there is precipitation falling as rain during the event.

The classification was performed by employing a cluster analysis technique to the hydrological and hydro-meteorological variables derived from E-HYPE simulation and observations for each of the identified flood events (See Table 1). Since, some of the employed variables could be correlated, we employed principal component analysis to derive variables that are

independent and have less dimensionality before we applied the clustering algorithm. We employed the k-means algorithm (Hartigan and Wong, 1979) with a large number of groups (20 groups as a starting point) and hierarchically merged groups using Ward's minimum variance method (Ward Jr., 1963). Two groups are merged in such a way that the increase in the total variance across all groups is the minimum. The distributions of the hydrological and hydro-meteorological variables within the identified clusters are used as a basis to establish the dominant flood generation mechanism for the events within a

given cluster member. After the automatic cluster analysis, the events in each group were carefully inspected and a manual adjustment was applied to move around events from one group to another if they happen to end up in a group which doesn't reasonably represent them.





### 2.6 Regional changes in flood types

Whether there have been changes in the occurrence of flood events and/or the dominant flood generating mechanisms in different regions of Europe was assessed through a significance test on the changes in the annual and seasonal numbers of different types of floods regionally. Since the number of events selected at a station level could be too few to enable

5 estimating a trend, we performed the test on the total regional counts of flood events. Five regions were defined based on the updated Köppen-Geiger climate classification (Peel et al., 2007), with some adjustments of the boundaries so that subcatchments delineated in the hydrological model employed in the study are not cut into different regions. The defined regions are: Alps, Northern, Western, Eastern, and Southern Europe (see Fig. 1). We employed the Mann-Kendal trend test (Kendall, 1975) on both the total number and the numbers of different types of flood events in each region. A change was

10 deemed significant at 5% significance level. The entire work procedure, together with the input used at each step, is schematized in Fig. 2.

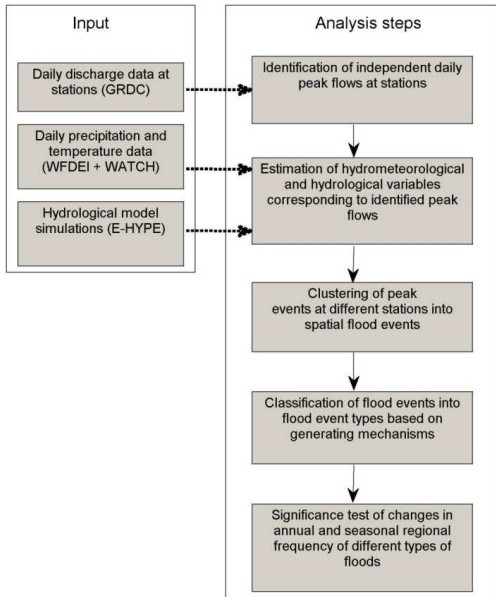

**Figure 2:** Schematic representation of the analysis procedure and the input data used in the study.





## 3 Results

### 3.1 Spatial clustering of flood events

Based on the analysis discussed in section 2.4, 3767 spatial flood events were identified across Europe over the period 1961-2010. Annually, the largest number of events occurred in the central part of Europe (Fig. 3) followed by regions in the

5    British Isles. Most parts of Northern Europe and Southern Europe, as well as parts of Europe further to the east displayed lower number of flood events. When the events were stratified seasonally, the highest number of winter events occurred in the Western and central parts of Europe as well as at several locations in the British Isles. A few number of events occurred elsewhere. Many of the summer events are concentrated in the Alps and parts of Northern and Eastern Europe. Overall, there were more events in winter than in summer (see Fig. 4).

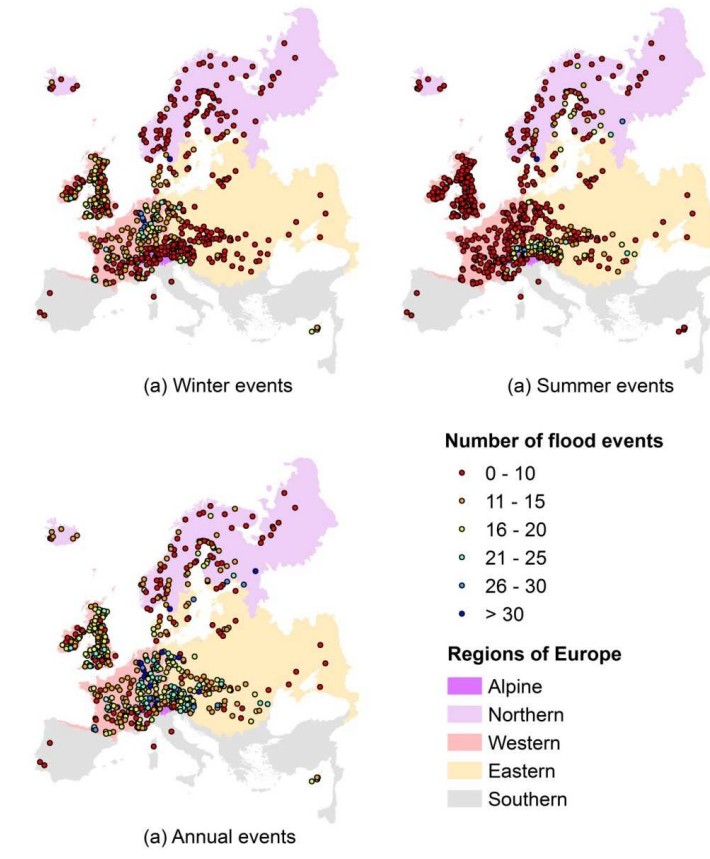

**Figure 3**: Total seasonal and annual counts of spatially clustered flood events each station is involved in over 1961-2010.





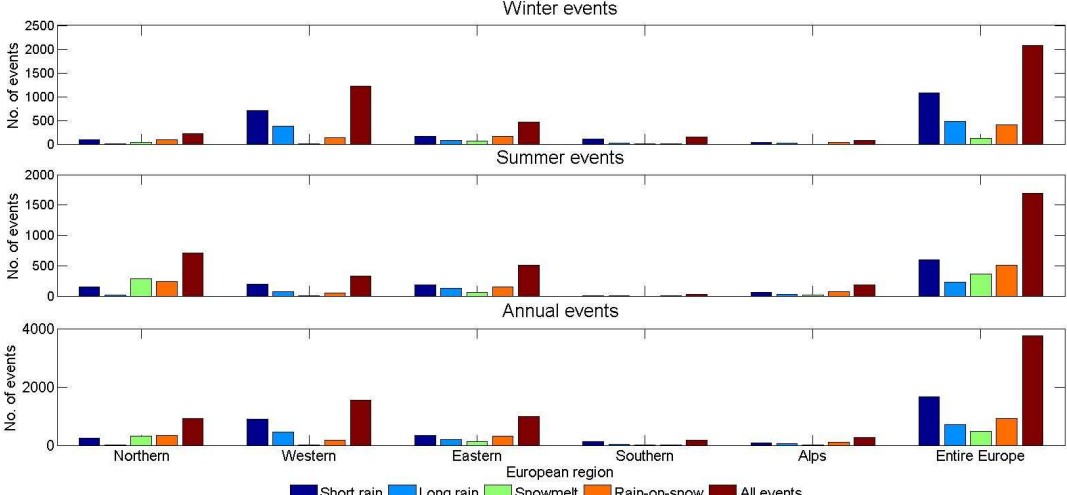

**Figure 4**: Total seasonal and annual counts of different types of flood events across Europe over 1961-2010.

### 3.2 Flood event types

As shown in Fig. 4, short rain flood types are dominant at the continental scale in all seasons when the identified spatially

clustered flood events were classified into the four types of flood events using the procedure outlined in Section 2.5.
Annually, this is followed by rain-on-snow events. Purely snowmelt events account for the least proportion of the total
events annually. Following short rain events, long rain events are the next dominant type of events in winter, while snowmelt
events are the least frequent ones. In summer, the most frequent event type next to short rain event is rain-on snow, followed
by snowmelt events. Long rain events are the least frequent in summer. Regionally, short rain events are dominant in all

regions only in winter. In summer, snowmelt or rain-on-snow events become dominant in northern Europe and the Alps.

As shown in Fig. 5, most stations in the western and southern parts of Europe as well as the southern part of the British Isles
and the southern tip of Scandinavia have a higher proportion of short rain flood events annually. Most parts of Central
Europe and northern part of the British Isles, as well as some parts of Western Europe have dominantly long rain flood
events. In the Northern part of Europe, either snowmelt or rain-on-snow events are dominant. In the upper part of the Alps,

most parts of the Rhine, as well as many parts of Eastern Europe, rain-on-snow events are dominant. Snow or rain-on-snow
events are little represented in western and southern Europe, as well as the British Isles.

Similar to the annual events, short rain events are dominant at most stations in western and southern Europe as well as the
southern part of the British Isles and the southern tip of Scandinavia in winter (see Fig. 6). In many parts of central Europe
and most parts of the British Isles, except the southern part, long rain events are dominant in winter. However, rain-on-snow

events are also important events in many parts of central and Eastern Europe in winter. There are few winter flood events in





Northern Europe and most of them are of rain-on-snow events with some purely snowmelt events, except in the south western part, where rainfall events are dominant.

In summer, Western Europe has either short rain or long rain events, with very little snowmelt or rain-on-snow events (Fig. 7). Events in southern Europe are few and those events are of short rain. In Northern Europe, either snowmelt or rain-on-snow events are dominant in summer. Short rain events are also important summer event types in some parts of Northern Europe. Long rain events are dominant in most parts of central Europe and some parts of Eastern Europe. In the Alps and the Rhine, as well as the Eastern part of Europe, rain-on-snow events are dominant summer event types.

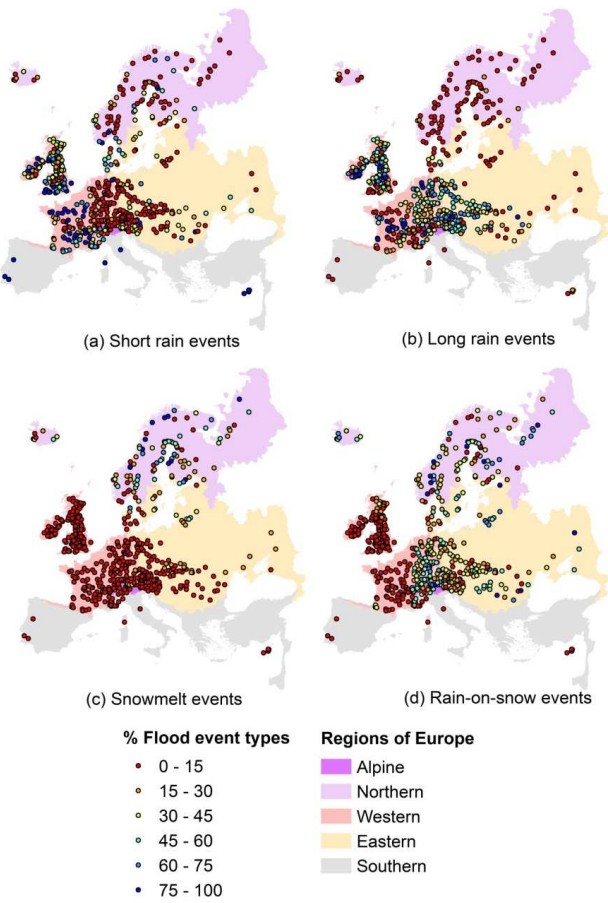

**Figure 5**: Percentages of different types of annual flood events in different regions of Europe over 1961-2010.





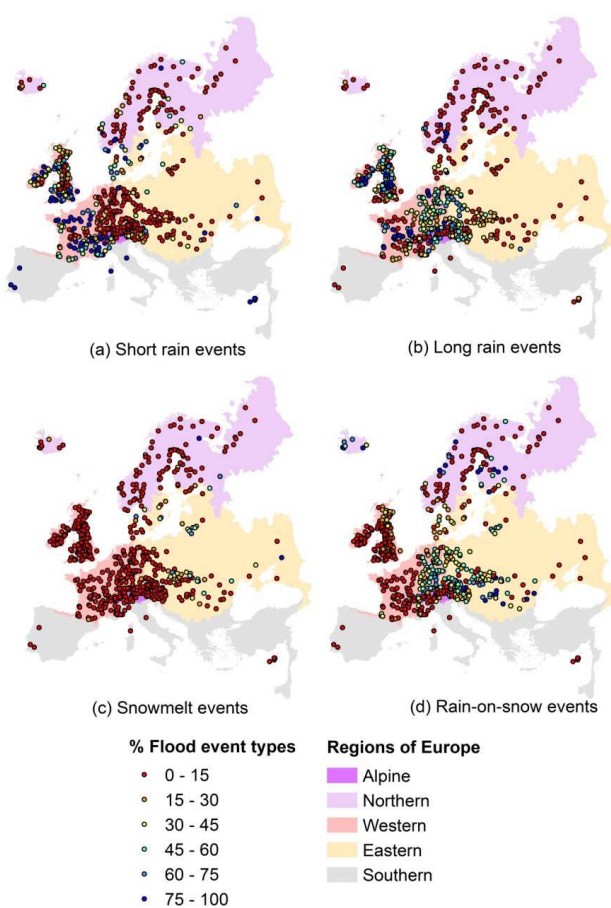

**Figure 6**: Same as Figure 5, but for winter flood events.

### 3.3 Hydrological and hydrometeorlogical event characteristics

Distribution of the different catchment states after the flood events were classified into different flood types (Fig. 8) shows

5    that the areal extent of short rain events, with a median value of 1,300 km², is the smallest compared to the other types of events. The median affected areal extents of all the other event types are comparable, which is around 7,000 km². However, rain-on-snow events have the largest variability, meaning that they could cover a wider range of areal extents when they occur. Purely snowmelt and long rain events have less range of variability of areal extent compared to rain-on-snow events but could cover a much wider range of areal extents than short rain events when they occur.



Distribution of the event duration of events displays a connection with whether snowmelt is involved in the process of event generation. Both snowmelt and rain-on-snow events show a higher median event duration than rainfall events (20 days versus 7 days). The variability of event duration is also higher for snowmelt and rain-on-snow events than for rainfall events. The range and variability is higher for rain-on-snow events compared to that of snowmelt events.

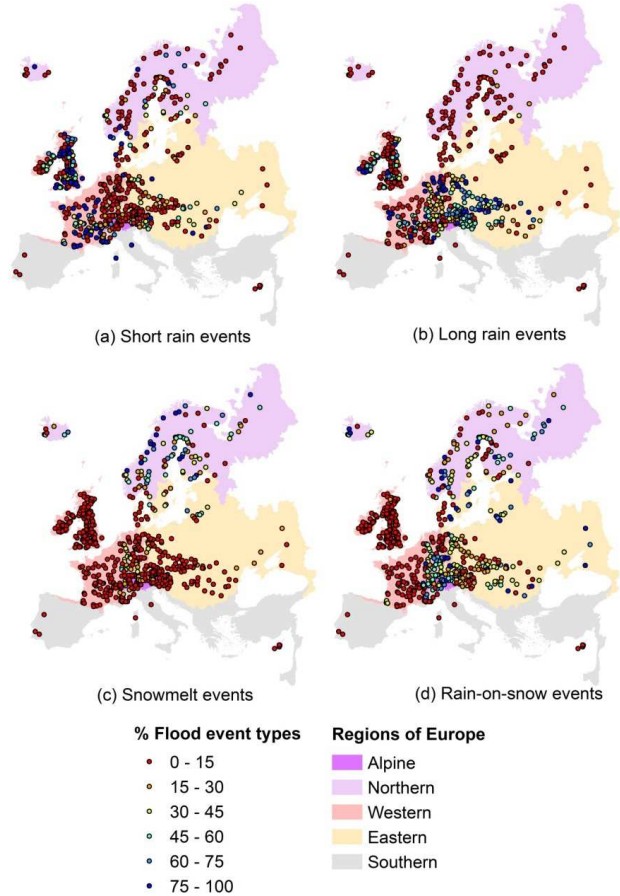

**Figure 7**: Same as Figure 5, but for summer flood events.

The median soil water moisture deficit at the onset of the flood events shows regional and seasonal variation. It is generally higher for summer events than winter events for all types of events. Regionally, it is higher in Eastern Europe and lower in the Alps, as shown in Table 2. The variability is higher for rainfall events compared to events with snow. On the other hand,




the antecedent precipitation index is generally higher for rainfall events than snowmelt and rain-on-snow events. It is especially the lowest for purely snowmelt events. A clear regional deference of antecedent precipitation is not discernible.

The amount of precipitation during a flood event is generally the highest for long rain events. For rain-on-snow events, the amount is higher than short rain events in most regions, especially in summer (Table 2). Regionally, event precipitation amount is generally higher in Southern Europe. This is especially the case in summer. In winter, the amount is also high in Northern Europe and the Alps.

Snow storage at the onset of events is generally higher for purely snowmelt than rain-on-snow events (Table 2 and Fig. 8). This is especially the case for summer events in regions where snow related events are important (Northern, Eastern, and the Alps). For purely snowmelt events in summer, the initial snow storage is the highest in Northern Europe and the Alps, while it is the highest in Eastern Europe for winter events. For rain-on-snow events, the highest median snow storage in both seasons was found for Southern Europe, followed by Northern Europe. However, as one can see in Table 2 and Fig. 4, there are very few rain-on-snow events in Southern Europe and the median amount of snow melt during the events is also low. The median amount of snowmelt during snowmelt and rain-on-snow events is the highest in Northern Europe, followed by Eastern Europe. In summer, it is the lowest in Western Europe, although the amount is comparable to that of Eastern Europe for winter events.

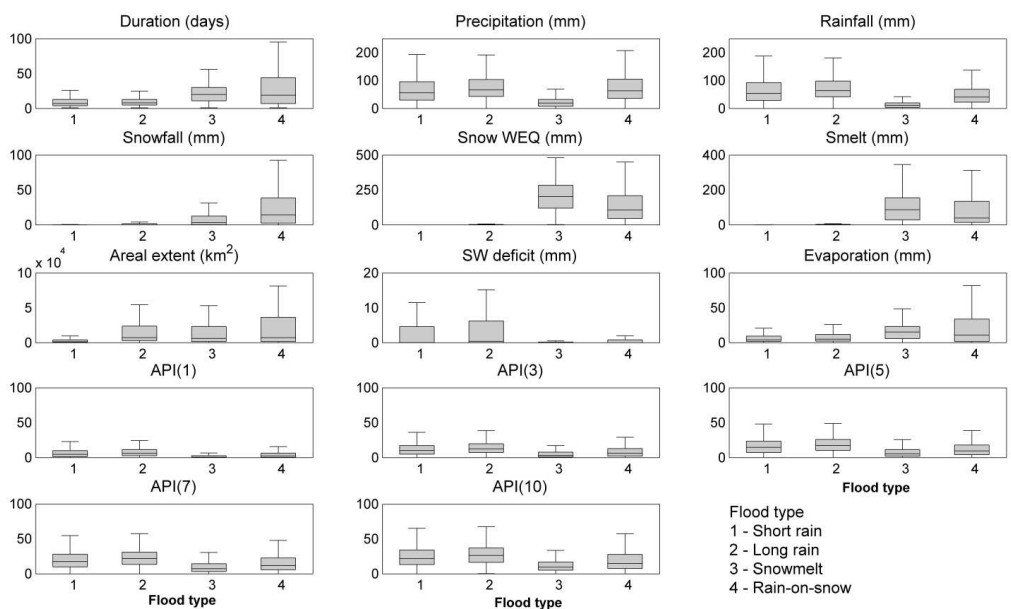

**Figure 8**: Distributions of different meteorological and hydrological state variables corresponding to the different types of flood events (See also Table 1 for description).



**Table 2**: Median values of seasonal and annual catchment event hydrometeorological characteristics corresponding to different food event types for different regions of Europe.

| Catchment states | Flood type | Northern Europe | | | Western Europe | | | Eastern Europe | | | Southern Europe | | | Alps | | | Entire Europe | | |
|---|---|---|---|---|---|---|---|---|---|---|---|---|---|---|---|---|---|---|---|
| | | Winter | Summer | Annual | Winter | Summer | Annual | Winter | Summer | Annual | Winter | Summer | Annual | Winter | Summer | Annual | Winter | Summer | Annual |
| Precipitation during event (mm) | SR | 64 | 40 | 55 | 58 | 46 | 55 | 55 | 51 | 52 | 79 | 89 | 79 | 85 | 47 | 53 | 60 | 47 | 56 |
| | LR | 137 | 39 | 56 | 66 | 56 | 65 | 89 | 59 | 70 | 116 | 117 | 116 | 117 | 55 | 86 | 72 | 58 | 66 |
| | SM | 7 | 22 | 20 | 16 | 14 | 14 | 21 | 17 | 18 | 0 | 0 | 0 | 0 | 15 | 15 | 17 | 20 | 20 |
| | RoS | 78 | 73 | 73 | 44 | 85 | 58 | 42 | 68 | 54 | 127 | 250 | 127 | 69 | 72 | 70 | 51 | 73 | 63 |
| Snow WEQ (mm) | SR | 2 | 0 | 0 | 0 | 0 | 0 | 0 | 0 | 0 | 0 | 0 | 0 | 0 | 0 | 0 | 0 | 0 | 0 |
| | LR | 16 | 0 | 0 | 0 | 0 | 0 | 3 | 0 | 0 | 0 | 0 | 0 | 1 | 0 | 0 | 0 | 0 | 0 |
| | SM | 91 | 258 | 244 | 78 | 60 | 60 | 111 | 162 | 133 | 0 | 0 | 0 | 0 | 260 | 260 | 97 | 236 | 202 |
| | RoS | 121 | 230 | 198 | 36 | 127 | 50 | 58 | 93 | 71 | 144 | 268 | 193 | 60 | 182 | 124 | 58 | 168 | 107 |
| Snow melt (mm) | SR | 3 | 0 | 0 | 0 | 0 | 0 | 1 | 0 | 0 | 0 | 0 | 0 | 1 | 0 | 0 | 0 | 0 | 0 |
| | LR | 14 | 0 | 3 | 0 | 0 | 0 | 6 | 0 | 0 | 0 | 0 | 0 | 3 | 0 | 1 | 0 | 0 | 0 |
| | SM | 5 | 136 | 121 | 23 | 32 | 23 | 26 | 84 | 41 | 0 | 0 | 0 | 0 | 46 | 46 | 20 | 115 | 88 |
| | RoS | 24 | 174 | 128 | 17 | 35 | 18 | 21 | 77 | 36 | 19 | 36 | 19 | 20 | 63 | 27 | 20 | 122 | 40 |
| Soil water deficit (mm) | SR | 0 | 1 | 0 | 0 | 2 | 0 | 0 | 12 | 4 | 0 | 7 | 0 | 0 | 0 | 0 | 0 | 3 | 0 |
| | LR | 0 | 4 | 3 | 0 | 4 | 0 | 1 | 16 | 8 | 1 | 3 | 1 | 0 | 2 | 1 | 0 | 8 | 0 |
| | SM | 0 | 0 | 0 | 0 | 2 | 0 | 0 | 0 | 0 | 0 | 0 | 0 | 0 | 0 | 0 | 0 | 0 | 0 |
| | RoS | 0 | 0 | 0 | 0 | 1 | 0 | 0 | 1 | 1 | 0 | 0 | 0 | 0 | 0 | 0 | 0 | 0 | 0 |
| API (1) | SR | 7 | 4 | 4 | 4 | 4 | 4 | 5 | 7 | 6 | 5 | 8 | 5 | 4 | 5 | 4 | 5 | 5 | 5 |
| | LR | 11 | 1 | 2 | 6 | 5 | 6 | 7 | 8 | 7 | 9 | 10 | 10 | 13 | 10 | 13 | 6 | 6 | 6 |
| | SM | 1 | 1 | 1 | 3 | 2 | 3 | 2 | 1 | 1 | 0 | 0 | 0 | 0 | 2 | 2 | 2 | 1 | 1 |
| | RoS | 4 | 1 | 1 | 5 | 4 | 5 | 3 | 2 | 2 | 8 | 7 | 7 | 8 | 3 | 4 | 4 | 1 | 2 |
| API (3) | SR | 14 | 11 | 12 | 10 | 10 | 10 | 10 | 12 | 11 | 10 | 8 | 10 | 11 | 9 | 10 | 10 | 11 | 10 |
| | LR | 32 | 5 | 9 | 12 | 12 | 12 | 14 | 17 | 16 | 14 | 22 | 18 | 18 | 23 | 21 | 12 | 14 | 13 |
| | SM | 9 | 3 | 3 | 17 | 15 | 16 | 5 | 4 | 5 | 0 | 0 | 0 | 0 | 5 | 5 | 7 | 3 | 3 |
| | RoS | 11 | 3 | 4 | 10 | 10 | 10 | 7 | 6 | 7 | 13 | 13 | 13 | 11 | 8 | 9 | 9 | 5 | 6 |

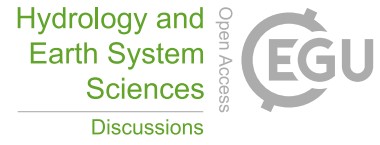

**Table 3:** Mann-Kendall significance levels of changes in the frequency of different types of flooding in different regions of Europe [signs in front of figures show direction of change, bold figures are significant at 5% level; SR=Short rain, LR=Long rain, SM=Snowmelt, RoS=Rain-on-snow, All= All events]

| | Northern Europe | | | | | Western Europe | | | | | Eastern Europe | | | | | Southern Europe | | | | | Alps | | | | | Entire Europe | | | | |
|---|---|---|---|---|---|---|---|---|---|---|---|---|---|---|---|---|---|---|---|---|---|---|---|---|---|---|---|---|---|---|
| | SR | LR | SM | RoS | All | SR | LR | SM | RoS | All | SR | LR | SM | RoS | All | SR | LR | SM | RoS | All | SR | LR | SM | RoS | All | SR | LR | SM | RoS | All |
| Winter | -46 | - | - | -54 | -17 | 2 | 1 | - | -30 | 2 | 2 | 53 | - | -8 | -96 | -3 | - | - | - | -1 | - | - | - | - | 99 | 15 | 1 | -7 | -12 | 40 |
| Summer | -16 | - | -36 | -4 | -3 | 66 | 4 | - | -27 | 34 | 69 | -44 | -12 | -3 | -26 | - | - | - | - | - | 3 | - | - | -26 | 47 | 91 | 90 | -17 | -1 | -9 |
| Annual | -7 | - | -15 | -14 | -3 | 3 | 1 | - | -45 | 2 | 51 | -72 | -44 | -5 | -28 | -7 | - | - | - | -1 | 5 | -87 | - | -61 | 52 | 17 | 1 | -6 | -2 | -74 |




### 3.4 Regional changes in flood types

As shown in Table 3, no significant changes are detected in the total number of events in all seasons at the continental scale. However, when the numbers of events were assessed based on the event types, the number of winter long rain events showed

5    a significant increase while the number of summer rain-on-snow events decreased significantly. Correspondingly, the annual numbers of long rain and rain-on-snow events showed significant positive and negative trends, respectively.

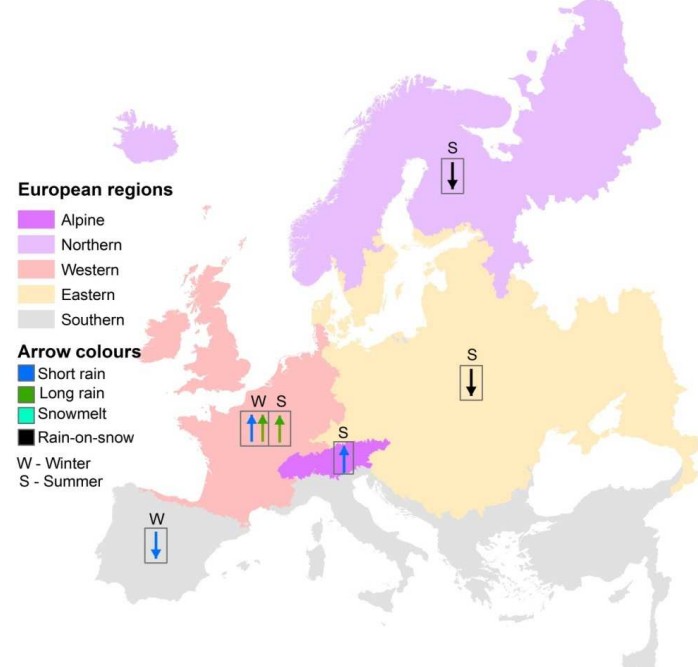

**Figure 9**: Regional distribution of significant seasonal changes in the frequency of different types of flood events across Europe (upwards arrows show significant positive trends and downward arrows show significant negative trends)

Regionally, the total annual number of events decreased significantly in Northern Europe, which is caused by a significant decrease in the number of summer rain-on-snow events (Fig. 9). No significant changes were detected in the numbers of the other types of flood in this region in all seasons. In Western Europe, the total annual number of events increased significantly and this was caused by a significant increase in the number of long rain events in both summer and winter, as well as number

15    of short rain events in winter. No significant changes were detected in the total annual number of events in Eastern Europe.



However, the annual number of rain-on-snow events decreased significantly, which was caused by a significant decrease in the number of summer rain-on-snow events. In winter, the number of short rain events showed a significant increase. Most of the events in the Alps occurred in summer and the dominant flood type is rain-on-snow. No significant change was detected in the number of this type of flooding in the Alps. However, the number of summer short rain events showed a

significant increase. The total annual number of events, however, didn't show any significant change.

Due to a few number of events in southern Europe as a result of very few stations, assessment of changes could only be done for winter short rain events, to which most of the identified events belong. The number of winter short rain events in this region identified based on those limited number of stations showed a significant decreasing trend.

**4 Discussion**

The regional distribution of the number of flood events and the frequency of different types of flooding can be put in perspective based on the hydro-meteorological features of each region and can also be compared with previous studies carried out locally for different regions. Most of the identified events, especially winter events were concentrated in the western and central parts of Europe and the event types were mainly rainfall events. Several studies in the past have associated winter flooding in these parts of Europe to extreme and often persistent precipitation events triggered by the

westerly winds bringing ample moisture from the Atlantic (Bárdossy and Caspary 1990; Beurton and Thieken 2009). Air temperature is milder in the western part of the region and snow is not an issue when it comes to major winter flooding, as can be seen in Fig. 6 by the absence of a significant number of snowmelt events in the western part. As one moves towards the east, however, the climate shifts to a continental climate and the air temperature may stay below freezing point, leading to accumulation of snow. As shown in Fig. 6, rain-on-snow events also become important event types in addition to purely

rainfall events towards the east. This happens often when the westerly winds bring rainfall together with a milder weather that leads to thawing of snow (Nied et al., 2015). Purely snowmelt events are also important events, although not as ubiquitous as rain-on-snow events. This finding highlights the more important role intense rainfall has in the winter flood genesis of western and central parts of Europe.

The summer events, which are mainly concentrated in the Alps and parts of Northern and central Eastern Europe, have a

more complex genesis, which are regionally variable. Although heavy rainfall is still an important mechanism, snow processes also become a significant contributor to the flood genesis in many parts. In the central eastern part of Europe, the majority of summer flood events are often caused by heavy rainfall that last for several days and are associated with moisture transport from the Adriatic Sea to central Europe over eastern Alps, which are often enhanced by orographic lifting (Jacobeit et al., 2006; Mudelsee et al., 2004; Ulbrich et al., 2003b, Jeneiová et al., 2016). The results shown in Fig. 7, where long rain

events appear to be the dominant summer event types in much of central Europe, confirm this. In addition, the colder winter leads to a longer retention of snow, which starts to melt in early or late spring, depending on the topographic feature (Beurton and Thieken 2009). Flooding can occur when this is enhanced by heavy rainfall (See Fig. 7). In Northern Europe,



melting of snow accumulated in the mountainous areas and regions further to the north during the cold winter control the summer flood regime (Arheimer and Lindström, 2015; Vormoor et al., 2015). This can be accompanied by rainfall, as shown in Fig. 7. Different flood generation mechanisms are associated with the flood regime in the Alps, which vary depending on the elevation and exposure. In the north of the Alps, a similar weather situation associated with heavy rainfall that lasts for

several days, which triggers most of the summer flooding in central Europe defines the flood regime (Böhm and Wetzel, 2006). A similar mechanism leads to flooding in the higher mountain ranges, but it can be accompanied by glacier and snowmelt, which enhance the antecedent catchment wetness (Merz and Blöschl, 2003; Parajka et al., 2010, Peña et al. 2015). A close examination of the different event characteristics corresponding to each of the flood types (Fig. 8) reveals the importance of catchment wetness at the onset of the event in addition to the meteorological forcing for flood generation. The

median soil moisture deficit at the onset of all types of events is low in most cases. One can also see that the variability of the deficit is higher for purely rainfall events than snow related events. The consistently low soil moisture deficit for snowmelt related events can be explained by the fact that snowmelt processes usually last for a longer period and have a tendency to saturate the soil before flooding ensues (see for instance Merz and Blöschl, 2003).  On the other hand, rainfall events may lead to flooding either through a saturation or infiltration excess process depending on the intensity of the rainfall. Flash

floods, for instance, often occur on relatively dry soil due to an infiltration excess process. The variability of the soil moisture deficit for rainfall events can be attributed to this variable mechanism of flood generation although the soil wetness is high for most of the events.

The higher flood event duration of snowmelt and rain-on-snow events in relation to purely rainfall events can also be attributed to the tendency of snowmelt processes to last longer and the corresponding wet antecedent catchment condition, as

discussed in the previous paragraph. Gaál et al (2012, 2015) and Szolgay et al. (2016) found similar results when they analyzed durations of flood events across Austria. They found that the flood event duration consistently increases with increasing snow-to-rainfall ratio of the process leading to the event despite the diversity of the non-climate catchment characteristics that have impact on the flood generation process that affect the flood event duration. They explained that as being a result of the wet antecedent condition related to snow processes, which leads to a higher volume of flood that

controls the flood duration. They also noted that for rain-on-snow events, even a moderate rainfall amount can lead to flooding since a higher base flow can be produced by snowmelt, thus leading to a diverse range of flood durations resulting from different amounts of rainfall. Their observation is in line with our result, which shows a higher variability of the duration of rain-on-snow flood events (see Fig. 8).

Some of the regionally and seasonally variable changes in the frequency of flooding obtained in this work can be compared

with the findings of past studies carried out regionally. For instance, the increasing trend in the occurrence of rainfall flood events in the western part of Europe, especially in winter is in line with studies conducted in the UK and Germany. Most studies reviewed by Hannaford (2015) suggest increasing frequency of winter flooding after the 60s. Based on the spatial coherence of the trends, the studies attributed the changes to climate related forcing, especially the increasing frequency of the westerlies that are linked to heavy precipitation events in most of Western Europe. Similarly, Petrow and Merz (2009)





found increasing frequency of winter flooding in the western, central and southern parts of Germany in the second half of the 20th century, which they speculated to be attributable to climate related forcing based on the spatial coherence of the changes. Hattermann et al (2013) confirmed this through analysis of corresponding changes in temperature and occurrence of heavy precipitation as well as atmospheric circulation patterns related to heavy precipitation.

In Northern Europe, on the other hand, past studies didn't show a clear and consistent pattern of change in the occurrence of flooding (Lindström and Bergström, 2004; Wilson et al., 2010). However, Wilson et al (2010) found that there is a clear tendency for early occurrence of spring floods which is likely due to an earlier snowmelt due to a rise in temperature. The declining trend in the frequency of summer rain-on-snow flood events in Northern and Eastern Europe found in our study can also be likely due to earlier snowmelt, which reduces the number of snow related events in the hydrological summer

season.

For Eastern and Central Europe, results from past studies show different patterns of change. Mudelsee et al. (2003) found a decreasing trend in the occurrence of winter flooding in the Oder and Elbe rivers while they found no significant trend in the occurrence of summer flooding over the past 80 to 150 years. They obtained similar trends in the occurrence of extreme precipitation. Petrow and Merz (2009), on the other hand, did not find any significant trend in the occurrence of extreme

flow in the Elbe river in both seasons over the 2nd half of the 20th century. Kundzewicz et al. (2013) found increasing trend in the number of occurrence of large flood events over 1985-2009 in large parts of Europe, with the majority of the events concentrated in Central and Eastern Europe. The time periods considered in the different studies are different, but no clear picture in the tendency of the change could be discerned. Our study shows an increasing tendency of the frequency of winter short rain flooding and a decline in the frequency of summer rain-on-snow events in the eastern part of Europe but no

significant change in the occurrence of flooding in general.

In the Alps, Schmocker-Fackel and Naef (2010) and Peña et al. (2015) found that the frequency of summer flooding in Switzerland has been on the rise in the last four decades and associated the changes to patterns of low-frequency atmospheric variability and corresponding changes in the frequency of heavy precipitation. Similar results are presented by Pekárová et al., (2016) who identified increasing frequency of large floods along the Danube river. Although our results are in agreement

with these studies in terms of the rise in the frequency of rainfall related summer flooding, we didn't find an increase in the frequency of summer flooding as snow related floods have decreased in frequency, offsetting the increase in rainfall related floods. Bard et al. (2012) found a trend to an earlier spring snowmelt in the Alps over a comparable time period. This could have led to less occurrence of snow related flooding in summer.

Only a few stations are selected in Southern Europe and they do not cover the entire region. Most of the events at the

available stations were winter short rainfall events and their frequency of occurrence showed a declining trend. Although it is difficult to obtain a greater picture of the changes in the flood regime due to a lack of good coverage of stations, the result obtained at the few stations looks to be in agreement with the findings of Mediero et al. (2014), who found decreasing trends in the occurrence of flood events in Spain for three different periods ending in 2009 and notably over the period 1959-2009.





Silva et al. (2012) also suggest a decline in the frequency of flooding in Portugal in recent decades, which they found to have a similar trend pattern with rainfall.

There are also a few European wide studies carried out on changes in flood regime using a consistent approach across the continent, whose results can be compared with ours. For instance, Mediero et al. (2015) carried out trend analyses on the magnitude, frequency, and timing of flooding at 102 discharge stations across Europe by dividing the continent into five regions based on flood seasonality. Their regions resemble, for the most part, the regions defined in our study. They found increasing trend in the frequency of flooding in the Northern Atlantic region, which covers the region defined as Western Europe in our study and decreasing trend in the Alps and regions defined as Southern Europe in our study over the period 1956-1995, which is closer to the time period of our study. Elsewhere, they didn't find a clear pattern of trend in the frequency of flooding. Their results appear to be in agreement with ours for Western and Southern Europe. Similarly, Mangini et al. (under review) studied trends in the magnitude and frequency of peaks-over-threshold flow series corresponding to different cross-over rates over 1965-2005 using daily discharge data sets at 629 GRDC stations for five hydro-climatic regions of Europe. They found increasing trends at several stations in Western and Northern Europe and decreasing trends at several stations in Eastern Europe and the Alps. Their results are in agreement with ours for Western Europe.

## 5 Conclusions

Spatiotemporal clustering of past flood events was performed using daily discharge data from rivers across Europe and the flood generation mechanism of each of the clustered events was identified. Four different types of flood generation mechanisms were defined: short rain, long rain, snowmelt, and rain-on-snow floods. The flood events were grouped into five different regions of Europe delineated based on the climate features of the regions. Annual and seasonal changes in the frequency of the different types of flooding were assessed for each region for the period 1961-2010.

Regional and seasonal differences were apparent in the frequency of occurrence of flood events. Most of the winter events occurred in western and central parts of Europe, which were mainly caused by rainfall events while the majority of summer events were concentrated in the northern and eastern parts of Europe as well as the Alps, where snowmelt related processes are important and most of the events are caused by either purely snowmelt or rain-on-snow processes.

At a continental scale, the frequency of winter short rain events increased significantly while frequency of summer rain-on-snow events deceased significantly. However, no significant change was detected in the frequency of occurrence of flooding in general. Regionally, significant decreasing trend was detected in northern Europe resulting from a decline in the frequency of occurrence of summer rain-on-snow events. Frequency of flooding increased significantly in Western Europe due to increasing frequency of rainfall events. Winter rainfall events increased significantly in frequency in Eastern Europe while summer rain-on-snow events decreased significantly. Overall, however, no significant change was detected in the frequency of flood occurrence. No significant change in the frequency of flooding was detected in the Alps although summer short rain





events showed a significant increase. Frequency of flooding in southern Europe decreased significantly due to a decline in rainfall events.

**Data availability**

Additional information on the experiment together with a protocol, data and codes is available at http://www.switch-on-vwsl.eu/.

The identified past spatial flood events, the corresponding hydrometeorological and hydrological characteristics, and the identified flood generating process types can be accessed at: https://doi.org/10.5281/zenodo.581454

Seasonal and annual regional changes in flood process types can be accessed at: https://doi.org/10.5281/zenodo.581452

**Acknowledgment**

This study was performed within the EU FP7-funded project SWITCH-ON [grant number 603587], which explores the potential of Open Data for comparative hydrology and collaborative research, as well as promote Open Science for transparency and reproducibility. The work was also supported by the ERC Advanced Grant "FloodChange" [number 291152, 2011]. All data, scripts and protocols used in the analyses are available in the SWITCH-ON Virtual Water-Science Laboratory at www.water-switch-on.eu.

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
