# Peer review of "Flood type classification and assessment of their past changes across Europe"

_Hydrology and Earth System Sciences, 2017_

## Referee Comment (RC1) · Anonymous Referee #1 · 24 Jul 2017

Review of "Flood type classification and assessment of their past changes across Europe"

This paper presents a broad assessment of floods in Europe by:

(i) gathering GRDC streamflow data from several hundred catchments in Europe. (ii) identifying floods events using base-flow separation techniques (iii) collecting hydrological and hydrometeorological state variables from existing data sets and models (iv) classifying floods into 4 different event types (short rainfall, multi day rainfall, rain on snow, snowmelt) (v) assessing changes in flood mechanisms and occurrences over time (vi) quantifying the spatial extend of flood. (vii) and presenting the links between iii-vii

From this analysis, the authors report extensively on all these aspects. While the presented results (iii-vii) are potentially a useful contribution to HESS, I cannot recommend publication of this work in HESS in its current format. Before I can recommend publication a large number of aspects need to be revised and clarified (which potentially will lead to a completely different paper). My main concerns are:

1) The paper is not explicit about the knowledge gap it fills. This needs to be emphasized much better. I provide specific comments on this in the detailed comments. But main concerns are:

- When reading the abstract I read a long list of results, but it is unclear what the novel take home message is. - When I read the introduction, I read many studies that have done partly overlapping work before, but when you state your aim (lines 1-2, page 4) it remains unclear what is really novel compared to earlier work. - When I read the methods section I have no idea what I should pay attention too, since I am not sure what the "new thing" is the paper will show me. - The results give a wide overview of "findings" but all of these are presented in a somewhat superficial manner (because you present so many different things) - The discussion section discusses some links with previous studies, but fails to reflect on what we really learn compared to earlier work. - The conclusions are just an overview of findings, not a conclusion about what we learned in this paper.

Better emphasizing the novel aspects of the paper is critical to understanding the contribution of your work.

2) The rationale of how flood events are defined is unclear and seems inappropriate to me; why is a flow where direct runoff (according to base flow separation as explained in section 2.2) that exceeds base flow and the mean flow, considered a flood? According to this definition a flow peak that just exceeds the mean annual flow, (and is mostly direct runoff) will be classified as a flood. However, in such a case flow conditions have nothing to do with flood conditions in my understanding. Of course, your method will

also likely identify real floods as flood events (since they exceed the mean annual flow, and are likely to have a lot of direct runoff) but these events will (probably) be much rarer than the events that I listed before. Therefor it seems that your study mostly characterizes events that should not be considered floods?

Without better justifying/clarifying this definition of floods the rest of the analysis has no value to me, since I have no idea if the presented statistics really characterize floods (or I am looking at some other flow characteristic)

3) The classification analysis is unclear, and seems inappropriate. Concerns are: - How can you define a classification scheme, and afterwards manually change for particular catchments to which class they belong? This approach is not repeatable and does not sound very scientific? Was your initial approach wrong anyway if you have to manually adjust afterwards? - All catchments are allocated to a particular class, but there are (potentially) important flood generating mechanisms that are not included in your four predefined flood types. For example, are soil moisture dynamics controlled by seasonal evaporation not important for floods in Europe? Are there any catchments where none of the posed mechanisms seem a reasonable explanation of the floods that you describe? - The description of the used cluster analysis is unclear (to me) (I cannot repeat the analysis presented at page 9 given on the provided information). In addition, the physical rationale why this clustering approach will actually lead to a reliable classification of flood mechanisms is unclear to me. Defining four mechanisms, and providing some clustering algorithm does not seem sufficient without explaining the physical rationale behind this approach (and emphasizing its limitations).

4) the method used to quantify the spatial extend of floods seems unreliable; first the chosen requirements are highly subjective (e.g. what would change in your analysis if you used different threshold conditions?). Second, these chosen conditions appear not to have to do anything with the definitions of flood outlined in section 2.2.? Third, the method seems highly sensitive to whether there is actually data available from nearby gauges available?

5) The presentation of all results needs to be improved. Currently a lot of vague terms are used to refer to results, without explicitly stating where (in e.g. the figure) the reader can see what result is referred to. An example would be "Distribution of the event duration of events displays a connection with whether snowmelt is involved in the process of event generation" (page 15, first lines) where unclear terminology is used (e.g. what do you mean by "shows a connection"?) and it is unclear where the reader can find these results (i.e. what figure do I need to look at, and what specific aspect within that figure?).

6) The writing of the paper needs to be improved a lot. While I provide a long list of suggestions in the detailed comments, this list is not exhaustive for the number of improvements that need to be made.

Considering these outlined concerns, I think the paper needs to be (extensively) revised before I can (consider) recommending this paper for publication in HESS.

Detailed comments

Page 1: Title: when I read the title, it remains unclear if "their" refers to "floods" or to the "flood classification". Consider rephrasing the title such that this is clear.

The abstract is a good overview of what is done in the paper, but it does not specify what is really new about the work, or what the specific hypothesis/niche is that this paper addresses. Explicitly including this in the abstract, will make it much easier for the reader to understand the novel contribution.

Line 10: "leading up to" or "causing"?

Line 10: later in the manuscript you state you use 614 catchments in the study (since you omitted the ones with too few data). 614 seems more appropriate to mention in the abstract than 745.

Line 11: "peak flows"? Can you be more specific since this is a poorly defined term.

Line 12: "form the same flood event" is unclear to me. Do you mean something like "are caused by the same driver"?

Line 13: "delineating" or something like ""providing a proxy for" (since it seems unlikely that you have enough data to really derive the spatial extend of flood events.

Line 14: what do you really mean by "are relevant in the flood generating process"?

Line 15: "for each of the identified spatially delineated flood events" Does this mean you first delineate the spatial extend of a flood, and after that for that "lumped" event search for the cause?

Line 13-15: "A pan-European . . . flood events". I have difficulty to really grasp what you try to say here. Maybe this is resolved by addressing the two previous comments

Line 16: "each flood event" does this refer to a single gauge or a group of catchments that together are part of the same bigger flood event?

Line 17: "were identified" is confusing. Make clear that these are 4 mechanisms originate from your modelling choices, and not from what data has taught you. Thus "tested" may be more appropriate than "identified".

Line 17: "long" and "short" are unclear. Is it not better to refer to as "(sub-)daily" and "multi-day"?

Line 17-18: "A trend . . . investigation period" is this trend analysis performed per mechanism (this giving a lumped picture of Europe) or per location (giving trends at individual catchments?)

Line 21: "did not" instead of "didn't"

Line 21: "total number of flood events" thus far it is unclear how you defined a flood. Consequently, I do not learn anything from this statement currently. The same problem applies to any of statements on flood changes in Lines 21-27.

Line 27: OK now I have read the entire abstract, but what is really the take home message? And what was the initial problem/niche that this paper addresses? Please clarify this to the reader (which is in line with my earlier comment that "The abstract is a good overview of what is done in the paper, but it does not specify what is really new about the work, or what the specific hypothesis/niche is that this paper addresses. Explicitly including this in the abstract, will make it much easier for the reader to understand the novel contribution")

Page 2: Line 3: "recent past"? Why not being more specific? For example, something like "past decade"?

Line 4: "The studies" or "These studies"?

Lines 4-5: "led to the events" or "caused these flood"?

Line 7: "there is a likelihood of their increase" is a meaningless statement; there is always some likelihood (it can just be bigger or smaller).

Lines 6-7: would it be useful to state WHY there is an increase in interest so this statement doesn't come out of the blue?

Line 8-10: It seems that all these studies characterize past changes, while you end the sentence before that talking about future flood changes; this may confuse the reader a bit.

Line 12: I am not sure if "proper" adds anything to this sentence? (or if it is even appropriate?)

Line 11-12: "Understanding the . . . local conditions". This argumentation is not complete; the argument needs to include a statement on how understanding of flood mechanisms helps flood management (which should be straightforward to include, and in general will help the reader to better understand the value of part of the work you present in the paper).

Line 15: do you mean "spatial" or "temporal" scales? Or both?

Line 16-20: this list of studies selected seems rather arbitrary. There are many more studies that characterize flood mechanisms? What is the value in listing these examples? More importantly, I am not interested in what "people have done before", I am interested in reading "what knowledge gap you are going to fill". In context of the latter statement, I suggest to reformulate such an overview to something like "while many studies characterized BLA BLA [references], it remains unclear STATE KNOWLEDGE GAP".

Lines 22-28: the same problem seems to apply here (but now for flood changes, rather than flood mechanisms classification)

Page 3 Lines 29 (page 2) – 16: the same problem seems to apply here (but now for flood change attribution)

Lines 17-33: the same problem seems to apply here

Page 4: "The aim … across Europe". This is an unclear goal to me, especially since there are other studies that do very related stuff already. Be more specific to make your goal clear.

Line 16: what screening criteria did apply that led to eliminating >100 catchments?

Line 17: which model did you use? (OK you will explain this later I see, but at a first read this confused me)

Line 18: Do you try to say you applied Hundecha's model results? (OK you will explain this later I see, but at a first read this confused me)

Line 20: How did you derive the HRU's? (OK you will explain this later I see, but at a first read this confused me)

Lines 27-28: Are these datasets somewhat consistent with another (or is combining them inappropriate for trend analysis?)

Section 2.1 What size catchments are we looking at?

Page 5:

Figure 1: Since you are inconsistent in the number of catchment that you use in this study (>745 in the abstract vs 614 in the methods) I do not know if this figure includes 747 or 614 catchments.

Figure 1: Where did you retrieve info that helped to delineate these different hydro-climatological regions? What is the rationale behind this classification? Why is that classification useful for your paper? It seems rather useful to include this information, since you use this classification later in the paper.

Line 4: "extreme flow" or "flood" (it may be useful to be consistent on wording throughout the manuscript)

Page 6:

Line 5 (page 4) – Line 20: I expressed my concern about the use of your "flood" definition in the main comments at the start of this review. In case you can make a decent rebuttal for this argument, please ensure that you much more clearly explain the rationale behind this definition? Are there other studies that use a similar definition of floods? What was their rationale behind choosing this definition?

Line 26: isn't every routine for land surface and subsurface processes "conceptual" at the scales we apply our models?

Page 7:

Line 3: "PET is achieved"? Don't you mean evaporation itself?

Most information on this page up to line 22 seems to come directly from Hundecha? Is it really worth repeating these details? Or can you make this section much shorter (and not confuse the reader since they are unsure whether you did this work, or just describe the data you borrowed from others?

Page 8:

Section 2.4: I do not see how these definitions (partly based on return periods) link to the definitions of flood peaks presented in section 2.2.

Page 9:

Line 17: what do you mean by "little"? This seems especially relevant since (i) this study should be repeatable and (ii) how much rain is needed to go from a snowmelt flood to a rain on snow flood?

Lines 30: Ok great you put effort in checking your results, but now that you "manually correct" some it seems like you just choose a wrong method to start with? Also, how can someone else repeat your analysis and results when you start changing results manually afterwards without specifying what the requirements are for you to change a catchment from one class to another?

Page 10: Ok, but what did you do with catchment where you manually changed the classification?

Page 11: Lines 9: Maybe it is worth to repeat the time periods you use to define "winter" and "summer"? (since people tend to skip to results).

Line 9: "more" can you be quantitative?

Figure 3: are there catchments where you identified no floods?

Figure 3: "Annual events" can be interpreted as "annual flood peaks" which are often used in flood studies. Maybe therefor change the label to avoid confusion?

Figure 3: Is there any solution to having so many catchments markers stacked on top of another in all the maps? It makes it difficult to see what is really going on in this region (same for the UK)

Figure 3: is it useful to add a frequency distribution of the number of recorded floods

over the stations?

Page 12: Figure 4: is it not much more useful to display the occurrence of flood types as percentages (or fractions) rather than total numbers? Now it is very difficult to see which processes are dominant in which region and how that varies between regions.

Line 6: "Annually, this is"? Please rephrase this.

Lines 9-10: "Regionally, short . . . in winter" I do not fully understand this sentence.

Line 16: "are little represented" or something like "rarely occur"?

Entire section 3.2: this description is too qualitative and vague to be useful to me. All these statements can be supported by for example including percentages or some other quantitative measure. For example, write things down like "short rain floods account for XX% of all recorded floods, and are thereby the dominant mechanism at the continental".

Page 14: "into different" or "according to"

Figures 5-7: How can you calculate these percentages when many of the catchment seem to have only a few floods recorded per catchment? Especially when you look at it per season (Figures 6 and 7)?

Section 3.3: Are these calculated areas not highly sensitive to the regional coverage of flow stations?

Line 7: variability in what?

Line 8: "less range of" or "smaller"

Page 15:

"displays a connection with whether snowmelt" such a formulation is really unclear to a reader. What connection do you see, which figure do we need to look at?

Discussion:

Reconsider "flood genesis" since it may be unclear what you mean (and will might confuse readers with a biblical reference).

Mangini et al. (under review): if this paper is not published by the time you revised your manuscript, I suggest removing this.

"didn't" or "did not"

I recommend rewriting this discussion section, in line with the main comment I provided at the start of this review.

Conclusions:

I recommend rewriting this conclusion, in line with the main comment I provided at the start of this review.

---

## Author Comment (AC1) · 8 Sep 2017

**Reply to interactive comment by Anonymous referee #1 on "Flood type classification and assessment of their past changes across Europe"**

**by Yeshewatesfa Hundecha, Juraj Parajka, Alberto Viglione**

This paper presents a broad assessment of floods in Europe by:
(i) gathering GRDC streamflow data from several hundred catchments in Europe. (ii) identifying floods events using base-flow separation techniques (iii) collecting hydrological and hydrometeorological state variables from existing data sets and models (iv) classifying floods into 4 different event types (short rainfall, multi day rainfall, rain on snow, snowmelt) (v) assessing changes in flood mechanisms and occurrences over time (vi) quantifying the spatial extend of flood. (vii) and presenting the links between iii-vii

From this analysis, the authors report extensively on all these aspects. While the presented results (iii-vii) are potentially a useful contribution to HESS, I cannot recommend publication of this work in HESS in its current format. Before I can recommend publication a large number of aspects need to be revised and clarified (which potentially will lead to a completely different paper). My main concerns are:
1) The paper is not explicit about the knowledge gap it fills. This needs to be emphasized much better. I provide specific comments on this in the detailed comments. But main concerns are:

- When reading the abstract I read a long list of results, but it is unclear what the novel take home message is. - When I read the introduction, I read many studies that have done partly overlapping work before, but when you state your aim (lines 1-2, page 4) it remains unclear what is really novel compared to earlier work. - When I read the methods section I have no idea what I should pay attention too, since I am not sure what the "new thing" is the paper will show me. - The results give a wide overview of "findings" but all of these are presented in a somewhat superficial manner (because you present so many different things) - The discussion section discusses some links with previous studies, but fails to reflect on what we really learn compared to earlier work. - The conclusions are just an overview of findings, not a conclusion about what we learned in this paper.
Better emphasizing the novel aspects of the paper is critical to understanding the contribution of your work.

*We thank the reviewer for the frank discussion and very thorough comments on the manuscript. We take the comments very seriously, particularly the parts where we need to be clearer in definitions and formulations of the novelty. The main objective of the paper is to describe past flood events in terms of their main flood generating mechanisms and hence flood types, and to investigate their regional patterns and changes in the frequency of occurrence across Europe over the past five decades. As the reviewer indicated, we have tried to review previous works done in assessing past changes in flooding, characterizing flood events in terms of processes, as well as those which attributed flood changes to changes in their drivers for different parts of Europe. However, this is, to our best knowledge, the first attempt to describe past flood events in terms of their generating mechanisms and assess their changes at the European scale. Some steps are involved in carrying out the work: identifying independent peak flows at stations, deriving the corresponding hydrological and hydrometeorological characteristics, identification of flood events based on certain definition of a flood event, describing the events in terms of the generating mechanisms, and assessing changes in the regional frequency of flood events generated by the different mechanisms. The methods section presents these steps in the order the work has been executed. Similarly, the results section presents the outcomes of the different steps and further goes into characterizing the different flood event types with respect to the distribution of their hydrological and hydrometeorological characteristics.*
*We will try to formulate the novel contribution and the messages more clearly.*

2) The rationale of how flood events are defined is unclear and seems inappropriate to me; why is a flow where direct runoff (according to base flow separation as explained in section 2.2) that exceeds base flow and the mean flow, considered a flood? According to this definition a flow peak that just exceeds the mean annual flow, (and is mostly direct runoff) will be classified as a flood. However, in such a case flow conditions have nothing to do with flood conditions in my understanding. Of course, your method will also likely identify real floods as flood events (since they exceed the mean annual flow, and are likely to have a lot of direct runoff) but these events will (probably) be much rarer than the events that I listed before. Therefor it seems that your study mostly characterizes events that should not be considered floods?

Without better justifying/clarifying this definition of floods the rest of the analysis has no value to me, since I have no idea if the presented statistics really characterize floods (or I am looking at some other flow characteristic)

*We thank the reviewer for this comment. The usage of inappropriate terminologies, such as flood events and extreme events in this section may have sent a wrong message. This section is actually meant to identify independent peak flows at each station used in the study. At this stage, they are not to be considered floods. The identified peak flows are used as a basis to sample from events based on different thresholds. Flood events are defined aggregating the peak flows in space and using thresholds, as described in Section 2.4. This could be better understood when looking at the Flow chart in Figure 2. We realize that the text and organization of the sections in the submitted manuscript may confuse the reader. In order to make things clearer, we will merge Sections 2.2 and 2.4 and describe them appropriately.*

3) The classification analysis is unclear, and seems inappropriate. Concerns are: -
How can you define a classification scheme, and afterwards manually change for particular catchments to which class they belong? This approach is not repeatable and does not sound very scientific? Was your initial approach wrong anyway if you have to manually adjust afterwards? - All catchments are allocated to a particular class, but there are (potentially) important flood generating mechanisms that are not included in your four predefined flood types. For example, are soil moisture dynamics controlled by seasonal evaporation not important for floods in Europe? Are there any catchments where none of the posed mechanisms seem a reasonable explanation of the floods that you describe? - The description of the used cluster analysis is unclear (to me) (I cannot repeat the analysis presented at page 9 given on the provided information). In addition, the physical rationale why this clustering approach will actually lead to a reliable classification of flood mechanisms is unclear to me. Defining four mechanisms, and providing some clustering algorithm does not seem sufficient without explaining the physical rationale behind this approach (and emphasizing its limitations).

*The clustering technique is employed to guide us into grouping the events in such a way that the resulting cluster groups have the desired distribution of the event hydrological and hydrometeorological characteristics based on our definition of the event types. We could have clustered the events based on all the event characteristics and tried to infer the characteristics of the resulting cluster groups. However, the resulting groups may not easily be defined in terms of the commonly employed flood process types. Therefore, we started by first defining the main flood generation mechanisms that can be identified based on the data we have. For some of the variables, it is difficult to define a clear border between the different mechanisms. For instance, what should be the rainfall amount to distinguish between snowmelt and rain-on-snow events? We performed clustering of the events based on their characteristics until we got groups whose statistical distributions of the event characteristics reasonably well describe the flood processes we defined rather than defining thresholds subjectively and grouping events based on such thresholds. This involves using different combinations of event characteristics in the clustering algorithm. The resulting groups have the desired statistical characteristics of the event characteristics but some of the individual events in a certain group may have event characteristics that are counterintuitive to the way the events are defined. For instance, an event with no snowmelt may end up in a group that represents snowmelt or rain-on-snow event. That is why we had to examine the events in each group and move around events accordingly. We used simple logical rules to move events around. This is an 'expert judgment' input to*

*the procedure which is hard to formalize using statistics only. The whole classification process could have been done manually by employing expert judgment based on the individual event characteristics, like in some of the works referred to in the manuscript such as Merz and Blöschl (2003) and others who employed similar procedures. However, the clustering algorithm we employed does much of the work for us and the manual adjustment refines the results. We will describe the last step of our procedure in more detail in the revised manuscript so that repeatability of the experiment will be possible.*

*We defined the flood generation mechanisms based on the commonly used definitions that are based on the forcing (Rainfall or snowmelt) .The effects of soil moisture are assessed within each group seasonally and regionally as discussed in section 3.3.*

*The clustering technique we employed is well established and we have made references to that. We can, however, give more details in the manuscript.*

4) the method used to quantify the spatial extend of floods seems unreliable; first the chosen requirements are highly subjective (e.g. what would change in your analysis if you used different threshold conditions?). Second, these chosen conditions appear not to have to do anything with the definitions of flood outlined in section 2.2.? Third, the method seems highly sensitive to whether there is actually data available from nearby gauges available?

*We agree that there are subjective elements in the spatial delineation of the flood events, as we have noted in Section 2.4. We defined flood events such that the events can potentially have impact. We used a minimum threshold of the 2 years flood as an approximation to the bankfull flow to delineate the flood extent and at least one location where a potentially impact causing flood level occurs. Different authors use different threshold for impact causing floods. Some use the 10 years flood and others the 5 years. The 10 year flood did not allow us to have enough events for change analysis. Therefore, we chose the 5 year flood as a threshold.*

*The peak flows discussed in section 2.2 were used as a basis to sample events from for the delineation (See the reply the second comment above).*

*The spatial coverage of the stations could affect the spatial delineation of the events. There is a possibility for events at different locations, which could have been classified as part of the same spatial event if more stations were available in the area, to be treated as separate events. But this is not something we have a fix to, at this stage, and assessing how sensitive the delineation is to the station density is beyond the scope of this paper.*

5) The presentation of all results needs to be improved. Currently a lot of vague terms are used to refer to results, without explicitly stating where (in e.g. the figure) the reader can see what result is referred to. An example would be "Distribution of the event duration of events displays a connection with whether snowmelt is involved in the process of event generation" (page 15, first lines) where unclear terminology is used (e.g. what do you mean by "shows a connection"?) and it is unclear where the reader can find these results (i.e. what figure do I need to look at, and what specific aspect within that figure?).

*Thank you for the suggestion. In response to this comment we will revise and add some sentences to more clearly formulate the results. In the particular example mentioned here, we started the section by making a reference to Figure 8, where most of the results discussed in the section are shown. The statement is meant to indicate that events where snowmelt is part of (snowmelt and rain-on-snow events) are associated with higher event duration compared to events without snowmelt (short rain and long rain events). But, yes we agree, it is important to make the formulations as clear as possible.*

6) The writing of the paper needs to be improved a lot. While I provide a long list of suggestions in the detailed comments, this list is not exhaustive for the number of improvements that need to be made. Considering these outlined concerns, I think the paper needs to be (extensively) revised before I can (consider) recommending this paper for publication in HESS.

*Thank you for the suggestions. We will carefully consider all of them.*

Detailed comments

Page 1: Title: when I read the title, it remains unclear if "their" refers to "floods" or to the "flood classification". Consider rephrasing the title such that this is clear.

*In response to this comment we will revise the title to avoid the ambiguity. In the current form, the 'their' refers to the flood types.*

The abstract is a good overview of what is done in the paper, but it does not specify what is really new about the work, or what the specific hypothesis/niche is that this paper addresses. Explicitly including this in the abstract, will make it much easier for the reader to understand the novel contribution.

*Thank you for the comment. We will add some statements in the abstract that emphasize the objectives of the paper.*

Line 10: "leading up to" or "causing"?

*Thank you for the suggestion. We will use a more appropriate word.*

Line 10: later in the manuscript you state you use 614 catchments in the study (since you omitted the ones with too few data). 614 seems more appropriate to mention in the abstract than 745.

*Thank you for the comment. We agree, and will fix the number in the abstract.*

Line 11: "peak flows"? Can you be more specific since this is a poorly defined term.

*'independent peak flows'. Please see the reply to the second general comment.*

Line 12: "form the same flood event" is unclear to me. Do you mean something like "are caused by the same driver"?

*Yes. That was the idea behind spatially delineating events. We will rewrite the statement to make it clearer.*

Line 13: "delineating" or something like ""providing a proxy for" (since it seems unlikely that you have enough data to really derive the spatial extend of flood events.

*Thank you for the suggestion.*

Line 14: what do you really mean by "are relevant in the flood generating process"?

*Thank you for the comment. '…relevant for the identification of flood generating processes'*

Line 15: "for each of the identified spatially delineated flood events" Does this mean you first delineate the spatial extend of a flood, and after that for that "lumped" event search for the cause?

*Yes. The spatially clustered events are the flood events we based our further analyses on.*

Line 13-15: "A pan-European : : : flood events". I have difficulty to really grasp what you try to say here. Maybe this is resolved by addressing the two previous comments

*Hopefully, it is clearer now.*

Line 16: "each flood event" does this refer to a single gauge or a group of catchments that together are part of the same bigger flood event?

*It refers to the latter.*

Line 17: "were identified" is confusing. Make clear that these are 4 mechanisms originate from your modelling choices, and not from what data has taught you. Thus "tested" may be more appropriate than "identified".

*Thank you for the suggestion. 'defined' would be more appropriate.*

Line 17: "long" and "short" are unclear. Is it not better to refer to as "(sub-)daily" and "multi-day"?

*We chose to use more general terms 'short' and 'long' since they often appears in the literature.*

Line 17-18: "A trend : : : investigation period" is this trend analysis performed per mechanism (this giving a lumped picture of Europe) or per location (giving trends at individual catchments?)

*Trend analysis was done both per mechanism and for all flood events. It was performed for each region and for the entire Europe as well. Details of the procedure are presented in Section 2.6.*

Line 21: "did not" instead of "didn't"

*Thank you for the suggestion.*
Line 21: "total number of flood events" thus far it is unclear how you defined a flood.
Consequently, I do not learn anything from this statement currently. The same problem applies to any of statements on flood changes in Lines 21-27.

*The flood events are the spatially clustered events described in Section 2.4. Please refer to our reply to the second general comment.*

Line 27: OK now I have read the entire abstract, but what is really the take home message?
And what was the initial problem/niche that this paper addresses? Please clarify this to the reader (which is in line with my earlier comment that "The abstract is a good overview of what is done in the paper, but it does not specify what is really new about the work, or what the specific hypothesis/niche is that this paper addresses. Explicitly including this in the abstract, will make it much easier for the reader to understand the novel contribution")

*Thank you again for the suggestion. We will try to emphasize the objectives of the work and put the findings in perspective. The main message of the paper is that we found a significant increase in the frequency of winter long rain events and a decrease in rain-on-snow events in the summer period. The results also show that there are (at the European scale) regional differences in the dominant flood generating mechanism and their changes over the investigation period.*

Page 2: Line 3: "recent past"? Why not being more specific? For example, something like "past decade"?

*Thanks for the suggestion. 'Past decades' would be more appropriate.*

Line 4: "The studies" or "These studies"?

*We meant to mention studies carried out to that effect in general not limited to those cited. But thank you for the suggestion.*

Lines 4-5: "led to the events" or "caused these flood"?

*Thank you for the comment. 'caused' would make the statement easier to comprehend.*

Line 7: "there is a likelihood of their increase" is a meaningless statement; there is always some likelihood (it can just be bigger or smaller).

*Thank you for the comment. We will reformulate the statement in a more understandable way.*

Lines 6-7: would it be useful to state WHY there is an increase in interest so this statement doesn't come out of the blue?

*The statement is a follow up to the previous statements. It is meant to indicate that interest has increased following the frequent occurrence of flood events. This is often mentioned in many of the works done recently, including many we referred. The statement will be reformulated accordingly.*
Line 8-10: It seems that all these studies characterize past changes, while you end the sentence before that talking about future flood changes; this may confuse the reader a bit.

*The referred study by Hall et al. (2014) actually reviews both past and projected future flood changes in Europe.*

Line 12: I am not sure if "proper" adds anything to this sentence? (or if it is even appropriate?)

*It was used to emphasize the importance of the knowledge on processes to get things done in the right way. But then, it is obvious and may be it is not relevant to emphasize.*

Line 11-12: "Understanding the : : : local conditions". This argumentation is not complete; the argument needs to include a statement on how understanding of flood mechanisms helps flood management (which should be straightforward to include, and in general will help the reader to better understand the value of part of the work you present in the paper).

*Thank you for the comment. We will add some statements on this.*

Line 15: do you mean "spatial" or "temporal" scales? Or both?

*We mean spatial scale. We will make it clear.*

Line 16-20: this list of studies selected seems rather arbitrary. There are many more studies that characterize flood mechanisms? What is the value in listing these examples?
More importantly, I am not interested in what "people have done before", I am interested in reading "what knowledge gap you are going to fill". In context of the latter statement, I suggest to reformulate such an overview to something like "while many studies characterized BLA BLA [references], it remains unclear STATE KNOWLEDGE GAP".

*Yes, we agree that the referred works are not exhaustive and as we indicated in line 6, they are just few examples among the many other similar works. Our aim was to indicate the different controls different authors worked with and we picked a few examples from each. A similar reasoning applies to the other aspects mentioned in the next three comments. After presenting a review of previous works done on the different aspects of flood process, classification, and flood changes, we state our objective and indicate where we are heading to in our work in the last paragraph of the introduction section. We agree that the last paragraph needs to be expanded in order to make it clear how our objective relates to the reviewed works and how it fills the knowledge gap.*

Lines 22-28: the same problem seems to apply here (but now for flood changes, rather than flood mechanisms classification)

*See the reply to the previous comment.*

Page 3 Lines 29 (page 2) – 16: the same problem seems to apply here (but now for flood change attribution)

*See the reply to the previous comment.*

Lines 17-33: the same problem seems to apply here

*See the reply to the previous comment.*

Page 4: "The aim : : : across Europe". This is an unclear goal to me, especially since there are other studies that do very related stuff already. Be more specific to make your goal clear.

*Thanks again. The paragraph will be expanded to make the objective clearer.*

Line 16: what screening criteria did apply that led to eliminating >100 catchments?

*Data must be available for at least 90% of the analysis period.*

Line 17: which model did you use? (OK you will explain this later I see, but at a first read this confused me)

*The section is on data employed in the study and the model was mentioned here only to indicate why we needed the data.*

Line 18: Do you try to say you applied Hundecha's model results? (OK you will explain this later I see, but at a first read this confused me)

*Same as above.*

Line 20: How did you derive the HRU's? (OK you will explain this later I see, but at a first read this confused me)

*Same as above.*

Lines 27-28: Are these datasets somewhat consistent with another (or is combining them inappropriate for trend analysis?)

*The data sets have some slight differences, as discussed in the reference we cited. Since we could not use only one of the data sets for the study period we had to combine them. We have evaluated all possible ways of combining the two data sets for possible inhomogeneities at the scale of the hydrological model resolution. We did not find any significant inhomogeneity with all combinations.*

Section 2.1 What size catchments are we looking at?

*The catchment sizes corresponding to the selected stations varies between 7 and 807,000km2.*

Page 5:
Figure 1: Since you are inconsistent in the number of catchment that you use in this study (>745 in the abstract vs 614 in the methods) I do not know if this figure includes 747 or 614 catchments.

*It shows the selected 614 stations. We will make it clear in the Caption.*

Figure 1: Where did you retrieve info that helped to delineate these different hydroclimatological regions? What is the rationale behind this classification? Why is that classification useful for your paper? It seems rather useful to include this information, since you use this classification later in the paper.

*We assess regional differences in the dominant flood types and their changes. Although the defined regions have geographical pattern, delineating the regions was not straightforward since there is no clear definition of the borders. Therefore, we augmented the delineation with a hydroclimatological map. The source is mentioned in Section 2.6. Yes, we agree, it appears in Figure 1 before we give any information on it. We will make a reference to Section 2.6 in the figure caption.*

Line 4: "extreme flow" or "flood" (it may be useful to be consistent on wording throughout the manuscript)

*'independent peak flows'. We will do corrections to the words accordingly. See also our reply to the second general comment.*

Page 6:
Line 5 (page 4) – Line 20: I expressed my concern about the use of your "flood" definition in the main comments at the start of this review. In case you can make a decent rebuttal for this argument, please ensure that you much more clearly explain the rationale behind this definition? Are there other studies that use a similar definition of floods? What was their rationale behind choosing this definition?

*See our reply to the second general comment.*

Line 26: isn't every routine for land surface and subsurface processes "conceptual" at the scales we apply our models?

*That depends on the employed model. The model we employed in our study has conceptual routines.*

Page 7:
Line 3: "PET is achieved"? Don't you mean evaporation itself?

*We have already defined the acronym in the statement preceding it and it should be ok since it is commonly used in the literature.*

Most information on this page up to line 22 seems to come directly from Hundecha?
Is it really worth repeating these details? Or can you make this section much shorter (and not confuse the reader since they are unsure whether you did this work, or just describe the data you borrowed from others?

*We included this description to indicate what degree of confidence we attach and the possible uncertainties to the model simulated variables used in the flood type classification. As a matter of fact, we added the paragraph at the editor's request.*

Page 8:
Section 2.4: I do not see how these definitions (partly based on return periods) link to the definitions of flood peaks presented in section 2.2.

*Please see our reply to the second general comment.*

Page 9:
Line 17: what do you mean by "little"? This seems especially relevant since (i) this study should be repeatable and (ii) how much rain is needed to go from a snowmelt flood to a rain on snow flood?

*Little refers to almost no rain. We didn't use a threshold value except that when there is no rainfall while there is snowmelt the event is regarded as a snowmelt event. As we mentioned in the reply to the third general comment, the final distribution of the variables after the cluster analysis was used to judge the classes.*

Lines 30: Ok great you put effort in checking your results, but now that you "manually correct" some it seems like you just choose a wrong method to start with? Also, how can someone else repeat your analysis and results when you start changing results manually afterwards without specifying what the requirements are for you to change a catchment from one class to another?

*Please have a look at the reply to the third general comment.*

Page 10: Ok, but what did you do with catchment where you manually changed the classification?

*See our reply to the third general comment.*

Page 11: Lines 9: Maybe it is worth to repeat the time periods you use to define "winter" and "summer"? (since people tend to skip to results).

*Thanks for the comment. Yes, it would be useful to repeat it once more here.*

Line 9: "more" can you be quantitative?

*Thanks for the comment. Yes, that can be included.*

Figure 3: are there catchments where you identified no floods?

*No, all catchments are part of at least a few events. Yes, the first class in the legend should change to <10 instead of 0 – 10. Thank you for the comment.*

Figure 3: "Annual events" can be interpreted as "annual flood peaks" which are often used in flood studies. Maybe therefor change the label to avoid confusion?

*Thank you for the comment. We actually mean all events without grouping them seasonally. 'All events' would be more appropriate.*

Figure 3: Is there any solution to having so many catchments markers stacked on top of another in all the maps? It makes it difficult to see what is really going on in this region (same for the UK)

*Removing the solid outline of the circles would make it a bit better*

Figure 3: is it useful to add a frequency distribution of the number of recorded floods over the stations?

*We believe so. It gives an idea on regions where more flood events have taken place.*

Page 12: Figure 4: is it not much more useful to display the occurrence of flood types as percentages (or fractions) rather than total numbers? Now it is very difficult to see which processes are dominant in which region and how that varies between regions.

*We are aiming to show the seasonal frequencies of the different flood types in each region and at the same time the regional differences in these frequencies. Usage of percentages would mask the regional differences in the frequencies.*

Line 6: "Annually, this is"? Please rephrase this.

*Thank you for the suggestion.*

Lines 9-10: "Regionally, short : : : in winter" I do not fully understand this sentence.

*We indicated that short rain events are dominant in all seasons at the continental scale. But when one looks at the regional differences in the dominant event type, short rain event is dominant in all regions only in winter.*

Line 16: "are little represented" or something like "rarely occur"?

*Thank you. That would be a better phrase.*

Entire section 3.2: this description is too qualitative and vague to be useful to me.
All these statements can be supported by for example including percentages or some other quantitative measure. For example, write things down like "short rain floods account for XX% of all recorded floods, and are thereby the dominant mechanism at the continental".

*Thank you for the suggestion. Point is well taken.*

Page 14: "into different" or "according to"

*'into the four defined flood types' could be more descriptive.*

Figures 5-7: How can you calculate these percentages when many of the catchment seem to have only a few floods recorded per catchment? Especially when you look at it per season (Figures 6 and 7)?

*It shows the proportion in percentage. Even if there were only event, it would be 100% of the class it belongs to and 0% for the other classes.*

Section 3.3: Are these calculated areas not highly sensitive to the regional coverage of flow stations?

*The results are based on the stations we have. Whether it changes with a different distribution of stations is not known but is possible. See the reply to the fourth general comment.*

Line 7: variability in what?

*Variability in the flood areal extent. That will be added in the statement. Thank you for the comment.*

Line 8: "less range of" or "smaller"

*Thank you for the suggestion.*

Page 15: "displays a connection with whether snowmelt" such a formulation is really unclear to a reader. What connection do you see, which figure do we need to look at?

*Thank you for the comment. See our reply to the fifth general comment as well.*

Discussion:
Reconsider "flood genesis" since it may be unclear what you mean (and will might confuse readers with a biblical reference).

*Thank you for the comment.*

Mangini et al. (under review): if this paper is not published by the time you revised your manuscript, I suggest removing this.

*We agree. We will remove it from the references if it is not going to be published.*

"didn't" or "did not"

*Thank you for the comment.*

I recommend rewriting this discussion section, in line with the main comment I provided at the start of this review.

*We will try to make corrections as appropriate.*

Conclusions:
I recommend rewriting this conclusion, in line with the main comment I provided at the start of this review.

*We will try to make corrections to this section as well as appropriate.*

---

## Referee Comment (RC2) · Anonymous Referee #2 · 28 Sep 2017

Assessments of flood change and potential classifications of floods across large regions, such as the continental scale analysis used here, are generally highly valuable contributions to the literature and science. Therefore, I believe the topic of the manuscript is relevant to HESS. However, the manuscript itself needs considerable revision to address some confusing parts of the methods, analysis, and presentation of the results. For this reason, I recommend the manuscript be reconsidered after major revisions and re-review.

Major comments:

1. Explanation of the Data and State Variables:

a. Section 2.1 could be better communicated as a table listing the variables and the

data source. The text can then be used to explain any additional pieces of information of particular note with the data, such as the fact that landuse and soils come from different sources (p. 4, lines 20-21). Also, the acronyms are not introduced before they are used.

b. I assume that the "hydrological model" being referred to on p. 4, line 17 is E-HYPE but that model is not named anywhere in this section. In Section 2.3, the model is discussed in more detail but this information is buried under a heading that indicates only the hydrologic and state variables are discussed. Consider renaming Section 2.3 and adding additional subsections to explicitly discuss the model, its input variables, and how the model is used in the analysis. These important pieces of the methods are currently not clear.

2. The clustering/classification by flood type is a main contribution of the manuscript. However, in Sections 2.4 and 2.5, the description of how the authors arrived at their flood classes needs substantial improvement. Here are examples of where I found these sections highly confusing:

a. Section 2.4 discusses the clustering of flood events and yet there are not details explaining how the clustering was quantitatively carried. No information is given as to how the stations were "clustered in space" p. 8, line 4. What variable was used to cluster? What method was used to cluster? Section 2.4 actually appears to be discussing how sites that may be exhibiting correlation in flood events due to their proximity to one another were filtered out - not anything about clustering of flood events.

b. Section 2.5 opens with a definition of 4 classes of flood events. This would indicate that the flood classes were determined a proiri and not by a formal clustering method. If the groups arise from a clustering algorithm, then I would consider them results and not appropriate to be placed in the Data and Methods section. As one reads further down however, there is information about a clustering method utilized but that includes the "hydrological and hydro-meterological variables defined from the E_HYPE model"

(p. 9, lines 23-24). In my comments above, I do not think enough information has been given about how the model variables are used in a clustering approach.

c. If the classes resulted from the application of a classification algorithm (in this case, the k-means algorithm), no evidence is given as to how the classification tree was pruned and how these classes were assigned a common behavior such as "short-rain floods." From the description of the flood types, it seems "short-rain floods" are defined as "a flood event caused by rainfall of duration less than one day" (p. 9, line 11). How was this definition arrived at - by looking at classified events to determine common properties or was this a pre-determined definition applied to the flood events.

d. Following on this, p. 9, lines 30-32 note that after the classification was complete, "manual adjustment" was used to move events around from group to another if they "happened to end up in a group which doesn't reasonably represent them." The authors need to provide objective criteria here as to how this was assessed. Since the remaining part of the manuscript centers around this classification, how can a reader be ensured the results are not biased by these initial adjustments? What was the point then of using a classification algorithm in the first place?

3. Because it is unclear how the flood classes were arrived at, the novelty of this work is not apparent. It would be more useful to pose the manuscript as a testing of several hypothesis about flood generating mechanisms or classes using the current state of the literature as support rather than general objective statements such as those found in p. 4, lines 1-4. I am not clear even as to whether the flood typing classes are a contribution because I am not sure if they are determined from the data or imposed by the authors to perform subsequent analysis.

4. Figures 3-7: These figures should be stand-alone. Referring back to previous captions decreases the readability and interpretation for the reader. I think it would also be helpful to show boxplots next to each map of the flood events grouped by region to show the distribution of the flood event types.

5. Figure 9: This figure is not well-explained and need clarification. Was a regional Kendall test used to obtain the significance values? If so, this is not cited or defined in the methods section. The conclusions made based on this analysis (p. 20, lines 1-8) do not reference any specific figure or evidence for these statements. This needs to be remedied.

6. Section 4: It is very difficult to follow some of the statements made in Section 4 (p. 21, lines 6-13 for example) because Figure 8 is so difficult to understand. I do commend the authors on using the discussion to pull together the literature on flood change and typing from smaller regions within Europe and describe how those studies fit with these results. I recognize synthesizing these results into the text was not a simple task.

Minor comments:

p. 4, line 15: The authors note a variable data period without mentioning how actually variable the periods are. Please note at least the minimum data period allowed.

p. 2, lines 22-24: This sentence is quite confusing as to what is meant here. The use of "different" twice creates most of the confusion.

Figure 8: Show the flood-type names instead of numerical values. This helps the reader to better understand the relation between the variables and the flood types.

---

## Author Comment (AC2) · 9 Oct 2017

**Reply to interactive comment by Anonymous referee #2 on "Flood type classification and assessment of their past changes across Europe"**

**by Yeshewatesfa Hundecha, Juraj Parajka, Alberto Viglione**

Assessments of flood change and potential classifications of floods across large regions, such as the continental scale analysis used here, are generally highly valuable contributions to the literature and science. Therefore, I believe the topic of the manuscript is relevant to HESS. However, the manuscript itself needs considerable revision to address some confusing parts of the methods, analysis, and presentation of the results. For this reason, I recommend the manuscript be reconsidered after major revisions and re-review.

*We thank the reviewer for the thorough review of the manuscript and thoughtful comments. We will attempt to clarify the issues raised.*

Major comments:

1. Explanation of the Data and State Variables:

a. Section 2.1 could be better communicated as a table listing the variables and the data source. The text can then be used to explain any additional pieces of information of particular note with the data, such as the fact that landuse and soils come from different sources (p. 4, lines 20-21). Also, the acronyms are not introduced before they are used.

*We thank the reviewer for the suggestion. In response to this comment, we will add a table that will summarize the datasets used in the study together with source links and references.*

b. I assume that the "hydrological model" being referred to on p. 4, line 17 is E-HYPE but that model is not named anywhere in this section. In Section 2.3, the model is discussed in more detail but this information is buried under a heading that indicates only the hydrologic and state variables are discussed. Consider renaming Section 2.3 and adding additional subsections to explicitly discuss the model, its input variables, and how the model is used in the analysis. These important pieces of the methods are currently not clear.

*Thank you for the comment. In response to this suggestion, we will rename Section 2.3. to "Hydrological modeling and flood event characteristics" and add an additional subsection that will describe the E-HYPE model and its implementation in the present work.*

2. The clustering/classification by flood type is a main contribution of the manuscript. However, in Sections 2.4 and 2.5, the description of how the authors arrived at their flood classes needs substantial improvement. Here are examples of where I found these sections highly confusing:

*Reviewer #1 has also raised points related to description of the clustering methodology. We will thus revise/describe the methodology in a more detailed way.*

a. Section 2.4 discusses the clustering of flood events and yet there are not details explaining how the clustering was quantitatively carried. No information is given as to how the stations were "clustered in space" p. 8, line 4. What variable was used to cluster? What method was used to cluster? Section 2.4 actually appears to be discussing how sites that may be exhibiting correlation in flood events due to their proximity to one another were filtered out - not anything about clustering of flood events.

*Grouping of the stations was performed based on peak events at the individual stations. Stations that are grouped into one spatially clustered event could be different for different spatially clustered events. A set of criteria were used to spatially group events at different stations. We defined flood events such that the events can potentially have impact. We set a minimum threshold of the 2 years flood at each station as an approximation to the bankfull flow to delineate the flood extent and put additional criterion that a potentially impact causing flood level occurs at least at one location. This is defined as the 5 years flood in our work. For each peak event exceeding the 5 year flood at each station, nearby stations with peaks exceeding the 2 year flood were searched for. Whether the stations fulfilling this criterion are grouped to form a spatially clustered event is decided based on the temporal lag of the peaks at the stations and the spatial distance between the catchments draining to the stations, as discussed in Section 2.4.*

b. Section 2.5 opens with a definition of 4 classes of flood events. This would indicate that the flood classes were determined a proiri and not by a formal clustering method. If the groups arise from a clustering algorithm, then I would consider them results and not appropriate to be placed in the Data and Methods section. As one reads further down however, there is information about a clustering method utilized but that includes the "hydrological and hydro-meterological variables defined from the E_HYPE model" (p. 9, lines 23-24). In my comments above, I do not think enough information has been given about how the model variables are used in a clustering approach.

*Yes, the classes were defined a priori. We have discussed this issue in a reply to a similar comment by reviewer #1 and we repeat it here. The clustering technique is employed to guide us into grouping the events in such a way that the resulting cluster groups have the desired distribution of the event hydrological and hydrometeorological characteristics based on our definition of the event types. We could have clustered the events based on all the event characteristics and tried to infer the classes from the characteristics of the resulting cluster groups. However, the resulting groups may not easily be defined in terms of the commonly employed flood process types. Therefore, we started by first defining the main flood generation mechanisms that can be identified based on the data we have. For some of the variables, it is difficult to define a clear border between the different mechanisms. For instance, what should be the rainfall amount to distinguish between snowmelt and rain-on-snow events? We performed clustering of the events based on their characteristics until we got groups whose statistical distributions of the event characteristics reasonably well describe the flood processes we defined rather than defining thresholds subjectively and grouping events based on such thresholds. This involves using different combinations of event characteristics in the clustering algorithm.*

c. If the classes resulted from the application of a classification algorithm (in this case, the k-means algorithm), no evidence is given as to how the classification tree was pruned and how these classes were assigned a common behavior such as "short-rain floods." From the description of the flood types, it seems "short-rain floods" are defined as "a flood event caused by rainfall of duration less than one day" (p. 9, line 11). How was this definition arrived at - by looking at classified events to determine common properties or was this a pre-determined definition applied to the flood events.

*Please refer to the reply to the previous comment.*

d. Following on this, p. 9, lines 30-32 note that after the classification was complete, "manual adjustment" was used to move events around from group to another if they "happened to end up in a group which doesn't reasonably represent them." The authors need to provide objective criteria here as to how this was assessed. Since the remaining part of the manuscript centers around this classification, how can a reader be ensured the results are not biased by these initial adjustments? What was the point then of using a classification algorithm in the first place?

*As we discussed in the reply to previous comments, the clustering technique was used as a guide to enable us classify events by avoiding subjective thresholds. The final cluster groups we arrived at have the desired statistical characteristics of the event characteristics based on the definitions but some of the individual events in a certain group may have event characteristics that are counterintuitive to the*

*way the events are defined. For instance, an event with no snowmelt may end up in a group that represents snowmelt or rain-on-snow event. That is why we had to examine the events in each group and move around events accordingly. We used simple logical rules to move events around. We will describe these rules in more detail in the revised manuscript.*

3. Because it is unclear how the flood classes were arrived at, the novelty of this work is not apparent. It would be more useful to pose the manuscript as a testing of several hypothesis about flood generating mechanisms or classes using the current state of the literature as support rather than general objective statements such as those found in p. 4, lines 1-4. I am not clear even as to whether the flood typing classes are a contribution because I am not sure if they are determined from the data or imposed by the authors to perform subsequent analysis.

*Based on our replies to the previous comments, we hope that our objective is clearer now. Our objective is to classify past flood events into flood types that are commonly discussed in the literature based on the characteristics of the meteorological drivers and hydrometeorological states across Europe and study the regional differences in the dominant flood generation mechanisms and the temporal trends. We didn't intend to introduce a novel methodology for flood type classification.*

4. Figures 3-7: These figures should be stand-alone. Referring back to previous captions decreases the readability and interpretation for the reader. I think it would also be helpful to show boxplots next to each map of the flood events grouped by region to show the distribution of the flood event types.

*Thank you for the suggestion. We will make the captions standalone in each figure and add distributions of the flood types in Figures 5-7.*

5. Figure 9: This figure is not well-explained and need clarification. Was a regional Kendall test used to obtain the significance values? If so, this is not cited or defined in the methods section. The conclusions made based on this analysis (p. 20, lines 1-8) do not reference any specific figure or evidence for these statements. This needs to be remedied.

*The Kendall test was performed on the regional counts of the different types of flood events and this was mentioned in section 2.6. The results of the trend test are presented in Table 3 and Figure 9 shows, for each region, the types of flood events that have shown significant trend. The conclusions on page 20 are based on figures shown in Table 3. We will make reference to the table.*

6. Section 4: It is very difficult to follow some of the statements made in Section 4 (p. 21, lines 6-13 for example) because Figure 8 is so difficult to understand. I do commend the authors on using the discussion to pull together the literature on flood change and typing from smaller regions within Europe and describe how those studies fit with these results. I recognize synthesizing these results into the text was not a simple task.

*Thanks for the comment. From line 8 on, it is actually a new paragraph and it is meant to discuss the characteristics of the different event types in terms of the meteorological and hydrometeorlogical characteristics. We will use subsections to enhance readability of the section.*

Minor comments:

p. 4, line 15: The authors note a variable data period without mentioning how actually variable the periods are. Please note at least the minimum data period allowed.

*The data obtained at the different stations are of different length. Some date as far back in time as 1812. The analysis in this work is based on a common period 1961-2010 and stations with at least 90% data over this period were used. We will add this information to the manuscript.*

p. 2, lines 22-24: This sentence is quite confusing as to what is meant here. The use of "different" twice creates most of the confusion.

*Thanks for the comment. We will reformulate the statement to "Studies have suggested that flood regime has been changing in Europe over the last decades, although the change pattern has been found to be regionally different".*

Figure 8: Show the flood-type names instead of numerical values. This helps the reader to better understand the relation between the variables and the flood types.

*Numbers were used not congest the figure area with texts as the names do not fit in an optimal way. We will show the flood types in a short form.*